# GAUSSIAN DIFFERENTIAL PRIVACY TRANSFORMATION: FROM IDENTIFICATION TO APPLICATION

## ABSTRACT

Gaussian differential privacy (GDP) is a single-parameter family of privacy notions that provides coherent guarantees to avoid the exposure of individuals from machine learning models. Relative to traditional $(\epsilon, \delta)$-differential privacy (DP), GDP is more interpretable and tightens the bounds given by standard DP composition theorems. In this paper, we start with an exact privacy profile characterization of $(\epsilon, \delta)$-DP and then define an efficient, tractable, and visualizable tool, called the Gaussian differential privacy transformation (GDPT). With theoretical property of the GDPT, we develop an easy-to-verify criterion to characterize and identify GDP algorithms. Based on our criterion, an algorithm is GDP if and only if an asymptotic condition on its privacy profile is met. By development of numerical properties of the GDPT, we give a method to narrow down possible values of an optimal privacy measurement $\mu$ with an arbitrarily small and quantifiable margin of error. As applications of our newly developed tools, we revisit some established $(\epsilon, \delta)$-DP algorithms and find that their utility can be improved. We additionally make a comparison between two single-parameter families of privacy notions, $\epsilon$-DP and $\mu$-GDP. Lastly, we use the GDPT to examine the effect of subsampling under the GDP framework.

## 1 INTRODUCTION

Recent years have seen explosive growth in the research and application of data-driven machine learning. While data fuels advancement in this unprecedented age of "big data", concern for individual privacy has deepened with the continued mining, transportation, and exchange of this new resource. While expressions of privacy concerns can be traced back as early as 1969 (Miller, 1969), the concept of privacy is often perceived as "vague and difficult to get into a right perspective" (Shils, 1966). Through its alluring convenience and promise of societal prosperity, the use of aggregated data has long outstripped the capabilities of privacy protection measures. Indeed, early privacy protection protocols relied on the ad hoc enforcement of anonymization and offered little to no protection against the exposure of individual data, as evidenced by the AOL search log and Netflix Challenge dataset controversies (Narayanan & Shmatikov, 2006; 2008; Barbaro & T. Zeller, 2006).

Differential privacy (DP) first gained traction as it met the urgent need for rigour and quantifiability in privacy protection (Dwork et al., 2006). In short, DP bounds the change in the distribution of outputs of a query made on a dataset under an alteration of one data point. The following definition formalizes this notion.

**Definition 1.1** *(Dwork et al., 2006) A randomized algorithm $\mathcal{A}$, taking a dataset consisting of individuals as its input, is $(\epsilon, \delta)$-differentially private if, for any pair of datasets $S$ and $S'$ that differ in the record of a single individual and any event $E$,*

$$P[\mathcal{A}(S) \in E] \leq e^\epsilon P\left[\mathcal{A}\left(S'\right) \in E\right] + \delta.$$

*When $\delta = 0$, $\mathcal{A}$ is called $\epsilon$-differentially private ($\epsilon$-DP).*

While the notion of $(\epsilon, \delta)$-DP has wide applications (Dankar & El Emam, 2012; Erlingsson et al., 2014; Cormode et al., 2018; Hassan et al., 2019), there are a few notable drawbacks to this framework. One is the poor interpretability of $(\epsilon, \delta)$-DP: unlike other concepts in machine learning, DP

should not remain a black box. Privacy guarantees are intended for human interpretation and so must be understandable by the users it affects and by regulatory entities. A second drawback is $(\epsilon, \delta)$-DP's inferior composition properties and lack of versatility. Here, "composition" refers to the ability for DP properties to be inherited when DP algorithms are combined and used as building blocks. As an example, the training of deep learning models involves gradient evaluations and weight updates: each of these steps can be treated as a building block. It is natural to expect that a DP learning algorithm can be built using differentially-private versions of these components. However, the DP composition properties cannot generally be well characterized within the framework of $(\epsilon, \delta)$-DP, leading to very loose composition theorems.

To overcome the drawbacks of $(\epsilon, \delta)$-DP, numerous variants have been developed, including the hypothesis-testing-based $f$-DP (Wasserman & Zhou, 2010; Dong et al., 2019), the moments-accountant-based Rényi DP (Mironov, 2017), as well as concentrated DP and its variants (Bun & Steinke, 2016; Bun et al., 2018). Despite their very different perspectives, all of these DP variants can be fully characterized by an infinite union of $(\epsilon, \delta)$-DP guarantees. In particular, there is a two-way embedding between $f$-DP and the infinite union of $(\epsilon, \delta)$-DP guarantees: any guarantee provided by an infinite union of $(\epsilon, \delta)$-DP can be fully characterized by $f$-DP and vice visa (Dong et al., 2019). Consequently, $f$-DP has the versatility to treat all of the above notions as special cases.

In addition to its versatility, $f$-DP is more interpretable than other DP paradigms because it considers privacy protection from an attacker's perspective. Under $f$-DP, an attacker is challenged with the hypothesis-testing problem

$$H_0 : \text{the underlying dataset is } S \quad \text{versus} \quad H_1 : \text{the underlying dataset is } S'$$

and given an output of an algorithm $\mathcal{A}$, where $S$ and $S'$ are neighbouring datasets. The harder this testing problem is, the less privacy leakage $\mathcal{A}$ has. To see this, consider the dilemma that the attacker is facing. The attacker must reject either $H_0$ or $H_1$ based on the given output of $\mathcal{A}$: this means the attacker must select a subset $R_0$ of $\mathrm{Range}(\mathcal{A})$ and reject $H_0$ if the sampled output is in $R_0$ (or must otherwise reject $H_1$). The attacker is more likely to incorrectly reject $H_0$ (in a type I error) when $R_0$ is large. Conversely, if $R_0$ is small, the attacker is more likely to incorrectly reject $H_1$ (in a type II error). We say that an algorithm $\mathcal{A}$ is $f$-DP if, for any $\alpha \in [0, 1]$, no attacker can simultaneously bound the probability of type I error below $\alpha$ and bound the probability of type II error below $f(\alpha)$. The function $f$ is called a trade-off function and controls the strength of the privacy protection.

The versatility afforded by $f$ can be unwieldy in practice. Although $f$-DP is capable of handling composition and can embed other notions of differential privacy, it is not convenient for representing safety levels as a curve amenable to human interpretation. Gaussian differential privacy (GDP), as a parametric family of $f$-DP guarantees, provides a balance between interpretability and versatility. GDP guarantees are parameterized by a single value $\mu$ and use the trade-off function $f(\alpha) = \Phi\left(\Phi^{-1}(1 - \alpha) - \mu\right)$, where $\Phi$ is the cumulative distribution function of the standard normal distribution. With this choice of $f$, the hypothesis-testing problem faced by the attacker is as hard as distinguishing between $N(0, 1)$ and $N(\mu, 1)$ on the basis of a single observation. Aside from its visual interpretation, GDP also has unique composition theorems: the composition of a $\mu_1$- and $\mu_2$-GDP algorithm is, as expected, $\sqrt{\mu_1^2 + \mu_2^2}$-GDP. This property can be easily generalized to $n$-fold composition. GDP also has has a special central limit theorem implying that all hypothesis-testing-based definitions of privacy converge to GDP in terms of a limit in the number of compositions. Readers are referred to Dong et al. (2019) for more information.

The benefits of DP come with a price. As outlined in the definition of DP, any DP algorithm must be randomized. This randomization is usually achieved by perturbing the intermediate step or the final output via the injection of random noise. Because of the noise, a DP algorithm cannot faithfully output the truth like its non-DP counterpart. To provide a higher level of privacy protection, a stronger utility compromise should be made. This leads to the paramount problem of the "privacy–utility trade-off". Under the $(\epsilon, \delta)$-DP framework, this trade-off is often characterized in a form of $\sigma = f(\epsilon, \delta)$: to achieve $(\epsilon, \delta)$-DP, the utility parameter (usually the scale of noise) needs to be chosen as $f(\epsilon, \delta)$. We will further discuss this formulation with examples in the next section.

## 1.1 OUR CONTRIBUTION

The general goal of this paper is to provide both a deeper theoretical understanding of and powerful practical tools for the GDP framework. To this end, we start with the analysis of privacy profiles

under the condition of traditional $(\epsilon, \delta)$-DP, where we point out an often-overlooked partial order on $(\epsilon, \delta)$-DP conditions via implication. We choose this as the starting point for our work because ignoring this partial order will lead to problematic asymptotic analysis and compromised utility.

Next, we broken down the GDP into two parts: a head condition and a tail condition and define an efficient, tractable, and visualizable tool, called the Gaussian differential privacy transformation (GDPT). We first used the GDPT on the identification of GDP algorithms. Through the intermediate property of GDPT that we develop, we find an easy-to-verify criterion that can distinguish GDP mechanisms from non-GDP mechanisms. For GDP algorithms, this criterion provides a lower bound for the privacy protection parameter $\mu$. This criterion can help researchers widen the set of available GDP algorithms and also gives an interesting characterization of GDP without an explicit reference to the Gaussian distribution.

Following the identification of a GDP algorithm, the logical next step is to measure the exact value of $\mu$. By development of numerical properties of the GDPT, we give a method to narrow down possible values of an optimal $\mu$ with an arbitrarily small and quantifiable margin of error.

Lastly, we give three more applications of our newly developed tools. In the first, we revisit some established $(\epsilon, \delta)$-DP algorithms and improve their utility by accounting for the overlooked partial order on $(\epsilon, \delta)$-DP conditions provided by logical implication. In the second application, we make a comparison between $\epsilon$-DP and $\mu$-GDP and find that any $\epsilon$-DP algorithm must be also $\mu$-GDP. In the last application, we discuss the effect of subsampling using the GDPT.

## 2 BACKGROUND

An algorithm can be $(\epsilon, \delta)$-DP for multiple pairs of $\epsilon$ and $\delta$: the union of all such pairs provides a complete image of the algorithm under the $(\epsilon, \delta)$-DP framework. In particular, an $(\epsilon, \delta)$-DP mechanism $\mathcal{A}$ is also $(\epsilon', \delta')$-DP for any $\epsilon' \geq \epsilon$ and any $\delta' \geq \delta$. The infinite union of $(\epsilon, \delta)$ pairs can thus be represented as the smallest $\delta$ associated with each $\epsilon$. This intuition is formulated as a privacy profile in Balle & Wang (2018). The privacy profile corresponding to a collection of $(\epsilon, \delta)$-DP guarantees $\Omega$ is defined as the curve in $[0, \infty) \times [0, 1]$ separating the space of privacy parameters into two regions, one of which contains exactly the pairs in $\Omega$. The privacy profile provides as much information as $\Omega$ itself. Many privacy guarantees and privacy notions, including $(\epsilon, \delta)$-DP, Rényi DP, $f$-DP, GDP, and concentrated DP, can be embedded into a family of privacy profile curves and fully characterized (Balle et al., 2020a). A privacy profile can be provided or derived by an algorithm's designer or users.

Before proceeding with detailed discussions, we first give three examples of DP algorithms that are used throughout the paper. The first example we consider is the Laplace mechanism, a classical DP mechanism whose prototype is discussed in the paper that originally defined the concept of differential privacy (Dwork et al., 2006). The level of privacy that the Laplace mechanism can provide is determined by the scale $b$ of the added Laplacian noise. Given a global sensitivity $\Delta$, the value of $b$ needs to be chosen as $f(\epsilon, 0) = \Delta/\epsilon$ in order to provide an $(\epsilon, 0)$-DP guarantee. Despite its long history, the Laplace mechanism has remained in use and study in recent years (Phan et al., 2017; Hu et al., 2019; Xu et al., 2020; Li & Clifton, 2021).

Our second example is a family of algorithms in which a noise parameter has the form $\sigma = A\epsilon^{-1}\sqrt{\log(B/\delta)}$. Examples include

- the goodness of fit algorithm (Gaboardi et al., 2016),
- noisy stochastic gradient descent and its variants (Bassily et al., 2014; Abadi et al., 2016; Feldman et al., 2018), and
- the one-shot spectral method and the one-shot Laplace algorithm (Qiao et al., 2021).

Our third example comes from the field of federated learning: given $n$ users and the number of messages $m$, the invisibility cloak encoder algorithm (ICEA) from (Ishai et al., 2006) is $(\epsilon, \delta)$-DP if $m > 10\log(n/(\epsilon\delta))$ (Ghazi et al., 2019). See also (Balle et al., 2020b; Ghazi et al., 2020) for other analysis of ICEA.

For figures and numerical demonstrations in this paper, we use $b = 2/\Delta$ for the Laplace mechanism; $A = 2$, $B = 1$, and $\sigma = 2$ for the second example, which we refer to as SGD; and $m = 20$ and $n = 4$

for the ICEA. We omit the internal details of these methods and focus on their privacy guarantees: other than for the classical Laplace mechanism, whose privacy profile is known (Balle et al., 2020a), privacy guarantees are given in the form of a privacy–utility trade-off equation $\sigma = f(\epsilon, \delta)$. Given $\sigma$, it is tempting to derive the privacy profile by inverting the trade-off equation (i.e., as $\delta_{\mathcal{A}}(\epsilon) = \min\{\delta \mid \sigma = f(\epsilon, \delta)\}$). However, in most cases, a privacy profile naively derived in this way is not tight and will lead to a problematic asymptotic analysis, especially near the origin, because of a frequently overlooked partial order between $(\epsilon, \delta)$-DP conditions, which we discuss below.

## 3 Privacy profiles and an exact partial order on $(\epsilon, \delta)$-DP conditions

An $(\epsilon_0, \delta_0)$-DP algorithm is trivially $(\epsilon, \delta)$-DP for any $\epsilon \geq \epsilon_0$ and $\delta \geq \delta_0$. However, this statement does not give a full picture of the relationships between different $(\epsilon, \delta)$-DP conditions.

**Theorem 3.1** *Assume that $\epsilon_0 \geq 0$ and $0 \leq \delta_0 < 1$. The $(\epsilon_0, \delta_0)$-DP condition implies $(\epsilon, \delta)$-DP if and only if $\delta \geq \delta_0 + (1 - \delta_0)(e^{\epsilon_0} - e^{\epsilon})^+/(1 + e^{\epsilon_0})$.*

**Corollary 3.1** *Assume that $\epsilon \geq \epsilon_0 \geq 0$ and $\delta \leq [(1 + e^{\epsilon})\delta_0 - e^{\epsilon} + e^{\epsilon_0}]/(1 + e_0^{\epsilon})$. An $(\epsilon, \delta)$-DP algorithm is $(\epsilon_0, \delta_0)$-DP.*

**Corollary 3.2** *Let $\mathcal{A}$ be an $(\epsilon_0, \delta_0)$-DP algorithm with the privacy profile $\delta_{\mathcal{A}}$. Then*

$$\delta_{\mathcal{A}}(\epsilon) \leq \delta_0 + \frac{(1 - \delta_0)(e^{\epsilon_0} - e^{\epsilon})^+}{1 + e^{\epsilon_0}}. \tag{1}$$

*We remark that the bound given in (1) is tight in the following sense: there is a specific $(\epsilon_0, \delta_0)$-DP algorithm $\mathcal{A}$ such that $\delta_{\mathcal{A}}(\epsilon)$ is exactly $\delta_0 + (1 - \delta_0)(e^{\epsilon_0} - e^{\epsilon})^+/(1 + e^{\epsilon_0})$.*

Corollary 3.1 states the exact partial order of logical implication on $(\epsilon, \delta)$-DP conditions. Taking this partial order into account, the privacy profile derived from the naive inversion of the trade-off function can be refined into

$$\delta_{\mathcal{A}}(\epsilon) = \min\left(\left\{\delta \mid \sigma = f(\epsilon_0, \delta_0) \text{ and } \delta \geq \delta_0 + \frac{(1 - \delta_0)(e^{\epsilon_0} - e^{\epsilon})^+}{1 + e^{\epsilon_0}}\right\}\right).$$

Intuitively, the refined privacy profile not only considers $(\epsilon, \delta)$-DP provided directly by the trade-off function but also takes all pairs $(\epsilon, \delta)$ inferred by Corollary 3.1. Theorem 3.2 can itself be used as a tool to improve some DP results: we will discuss this improvement in Section 5. With this approach, we can derive the privacy profile of our second example to be $\delta_{\text{SGD}}(\epsilon) = \min(\delta_2 + (1 - \delta_2)(e^{\epsilon_2} - e^{\epsilon})^+/(1 + e^{\epsilon_2}), \exp\{-\epsilon^2\})$, where $\epsilon_2 \approx 1.187$ and $\delta_2 \approx 0.244$. The privacy profile of our third example is $\delta_{\text{ICEA}}(\epsilon) = \min(\delta_3 + (1 - \delta_3)(e^{\epsilon_3} - e^{\epsilon})^+/(1 + e^{\epsilon_3}), K/\epsilon)$, where $K = 4/e^2$, $\epsilon_3 \approx 1.159$, and $\delta_3 \approx 0.468$. Notice that $\delta_{\text{SGD}}$ and $\delta_{\text{ICEA}}$ share a similar form: both are the smaller of two terms, one derived from Corollary 3.1 from a particular $(\epsilon, \delta)$-DP pair and the other from inverting the trade-off function. This is no coincidence. We leave the derivation to Appendix B.

We next show the connection between GDP and the privacy profile: briefly, Gaussian differential privacy can be characterized as an infinite union of $(\epsilon, \delta)$-DP conditions.

**Theorem 3.2** *A mechanism is $\mu$-GDP if and only if it is $(\epsilon, \delta_\mu(\epsilon))$-DP for all $\epsilon \geq 0$, where*

$$\delta_\mu(\epsilon) = \Phi\left(-\frac{\epsilon}{\mu} + \frac{\mu}{2}\right) - e^{\epsilon}\Phi\left(-\frac{\epsilon}{\mu} - \frac{\mu}{2}\right). \tag{2}$$

This result follows from properties of $f$-DP and is formulated as Corollary 2.13 in Dong et al. (2019). Prior to this general form, a similar expression for a special case appeared in Balle & Wang (2018). From the definition of the privacy profile, it follows immediately that an algorithm $\mathcal{A}$ with the privacy profile $\delta_{\mathcal{A}}$ is $\mu$-GDP if and only if $\delta_\mu(\epsilon) \geq \delta_{\mathcal{A}}(\epsilon)$ for all non-negative $\epsilon$. However, this observation does not automatically lead to a meaningful way to identify $\mu$-GDP algorithms.

Before proceeding with an analysis of privacy profiles, we give a few visual examples in Figure 1. The left pane illustrates the privacy profiles of our examples. That of the Laplace mechanism

is derived in Balle et al. (2020a) as Theorem 3: given a noise parameter $b$ and a global sensitivity $\Delta$, the privacy profile of the Laplace mechanism is $\delta(\epsilon) = \max(1 - \exp\{\varepsilon/2 - \Delta/(2b)\},\ 0)$. For the second and the third examples, we compare the naive privacy profiles obtained by inverting the trade-off function and the refined privacy profiles. The refined and naive privacy profiles take on notably different values around $\epsilon = 0$. The inverted trade-off functions suggest that $(0, \delta)$ cannot be achieved by any choice of parameter $\sigma$. However, this is clearly not true, considering Theorem 3.1.

As shown in the right pane of Figure 1, the Laplace mechanism's privacy profile is below the 2-GDP and 4-GDP curves but crosses the 1-GDP curve, indicating that the Laplace mechanism in this case is 2-GDP and 4-GDP but not 1-GDP. The ICEA curve intersects all of the displayed GDP curves, so the algorithm is not $\mu$-GDP for $\mu \in \{1, 2, 4\}$. It is hard to tell whether or not the SGD curve crosses the 1-GDP curve and we cannot say if it will cross the 2-GDP or even the 4-GDP curve at a large value of $\epsilon$. These examples illustrate that we cannot draw conclusions simply by looking at a graph. A privacy profile is defined on $[0, \infty)$, so it is hard to tell if an inequality is maintained as $\epsilon$ increases. Previous failures of ad hoc attempts at privacy have taught that privacy must be protected via tractable and objective means (Narayanan & Shmatikov, 2006; 2008; Barbaro & T. Zeller, 2006).

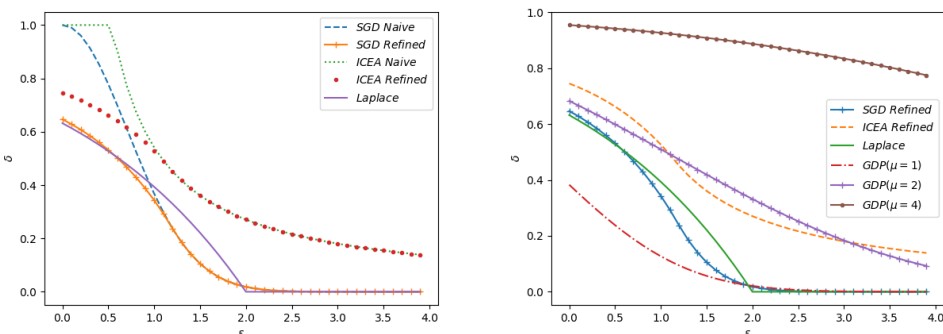

Figure 1: (Left) Examples of privacy profiles obtained by inverting the trade-off function (naive) and by Theorem 3.1 (refined). (Right) Comparison of 1-GDP and 2-GDP privacy profiles against those for our three examples.

Performing this check via numerical evaluation yields similar problems: we cannot consider all values of $\epsilon$ on an infinite interval (or even a finite one, for that matter). Turning to closed forms for privacy profiles and $\delta_\mu$ is also difficult: even if a given privacy profile is easy to handle, $\delta_\mu$ presents some technical hurdles. The profile $\delta_\mu$ and $\Phi$ are transcendental with different asymptotic behaviors for different values of $\mu$ and $\epsilon$. This is clear from the right pane of Figure 1: near $\epsilon = 0$, $\delta_\mu$ is concave for $\mu = 4$ but convex for $\mu = 1$. As a further complication, both the first and second terms in the definition of $\delta_\mu$ converge to 1 as $\epsilon \to \infty$, but the difference between them vanishes. Subtracting good approximations of two nearby numbers may cause a phenomenon called catastrophic cancellation and lead to very bad approximations (Malcolm, 1971; Cuyt et al., 2001). Due to the risk of catastrophic cancellation, a good approximation of $\Phi$ does not guarantee a good approximation of the GDP privacy profile. These problems make it difficult to tightly bound $\delta_\mu$ by a function with a simple form.

To address the problem of differing asymptotic behaviours, we define the following two notions.

**Definition 3.1** *(Head condition) An algorithm $\mathcal{A}$ with the privacy profile $\delta_\mathcal{A}$ is $(\epsilon_h, \mu)$-head GDP if and only if $\delta_\mathcal{A}(\epsilon) \le \delta_\mu(\epsilon)$ when $\epsilon \le \epsilon_h$.*

**Definition 3.2** *(Tail condition) An algorithm $\mathcal{A}$ with the privacy profile $\delta_\mathcal{A}$ is $(\epsilon_t, \mu)$-tail GDP if and only if $\delta_\mathcal{A}(\epsilon) \le \delta_\mu(\epsilon)$ when $\epsilon > \epsilon_t$. In particular, we define $\mathcal{A}$ as $(+\infty, \mu)$-tail GDP if $\mathcal{A}$ is $(\epsilon_t, \mu)$-tail GDP for some $\epsilon_t < +\infty$ and define $\mathcal{A}$ as $(\epsilon_t, +\infty)$-tail GDP if $\mathcal{A}$ is $(\epsilon_t, \mu)$ for some $\mu < +\infty$.*

The head condition checks the $\mu$-GDP condition for $\epsilon$ near zero and the tail condition checks the $\mu$-GDP condition for $\epsilon$ far away from zero. As such, the combination of $(\epsilon, \mu)$-head GDP and $(\epsilon, \mu)$-tail GDP is equivalent to $\mu$-GDP.

## 4  THE GAUSSIAN DIFFERENTIAL PRIVACY TRANSFORMATION

In this section, we propose a new tool called the Gaussian Differential privacy transformation (GDPT). We start by establishing a link between the GDPT and the head and tail conditions and discuss how the GDPT can identify GDP algorithms and measure the value of $\mu$.

**Definition 4.1** *(GDPT) Let $f$ be a non-increasing, non-negative function defined on $[0, +\infty)$ satisfying $f(0) \leq 1$. The Gaussian differential privacy transformation (GDPT) of $f$ is the function $G_f$ mapping $[0, \infty)$ to $[0, \infty)$ such that $G_f(\epsilon) = \mu_{GDP}(\epsilon, f(\epsilon))$, where $\mu_{GDP}(x, y)$ is the implicit function defined by the equation $\delta_\mu(x) = y$.*

We highlight two critical features of the GDPT.

- The GDPT is order preserving: if $f(\epsilon) \geq g(\epsilon)$, then $G_f(\epsilon) \geq G_g(\epsilon)$.
- The GDPT of $\delta_\mu$ is $G_{\delta_\mu}(\epsilon) = \mu$, a constant function.

The first of these two features derives from the monotonicity of $\delta_\mu(\epsilon)$. Given a fixed $\mu$, $\delta_\mu(\epsilon)$ is a strictly decreasing continuous function of $\epsilon$. Given a fixed $\epsilon$, $\delta_\mu(\epsilon)$ is a strictly increasing continuous function of $\mu$. Therefore, $\mu_{GDP}(x, y)$ is an increasing function of $y$: this leads to the order-preserving property. The second property follows immediately from the definition of $\mu_{GDP}$.

By taking advantage of the order-preserving property, direct comparisons between $\delta_\mu$ and $\delta_\mathcal{A}$ are no longer necessary: instead, it is sufficient to compare their corresponding GDPTs. Furthermore, appealing to the second property above, we need only compare $G_{\delta_\mathcal{A}}$ to the constant function $\mu$. The following theorems formalize this insight.

**Theorem 4.1** *An algorithm $\mathcal{A}$ with the privacy profile $\delta_\mathcal{A}$ is $(\epsilon_h, \mu)$-head GDP if and only if $\mu \geq \sup(\{G_{\delta_\mathcal{A}}(\epsilon) \mid \epsilon \in [0, \epsilon_h])$.*

**Theorem 4.2** *An algorithm $\mathcal{A}$ with the privacy profile $\delta_\mathcal{A}$ is $(\epsilon_t, \mu)$-tail GDP if and only if $\mu \geq \sup(\{G_{\delta_\mathcal{A}}(\epsilon) \mid \epsilon \in (\epsilon_t, \infty))$.*

**Corollary 4.1** *An algorithm $\mathcal{A}$ with the privacy profile $\delta_\mathcal{A}$ is $\mu$-GDP if and only if $\mu \geq \sup(\{G_{\delta_\mathcal{A}}(\epsilon) \mid \epsilon \in [0, \infty)\})$.*

Without the above results, we would be forced to search through an uncountably-large family of functions for a single $\delta_\mu$ that never crosses $\delta_\mathcal{A}$ anywhere on $[0, \infty)$ and has $\mu$ as small as possible. Now, with Theorem 4.1, we need only consider one function: the GDPT of $\delta_\mathcal{A}$. The tightest value $\mu$ is $\sup_\epsilon \{G_{\delta_\mathcal{A}}(\epsilon)\}$.

Using the GDPT framework, a condition for identifying GDP algorithms (that does not specify the exact value of $\mu$) is simple to formulate. We do so in the following theorems.

**Theorem 4.3** *An algorithm $\mathcal{A}$ is GDP if and only if $\mathcal{A}$ is $(+\infty, +\infty)$-tail GDP.*

**Theorem 4.4** *Let $f$ be a non-increasing, non-negative function defined on $[0, +\infty)$ satisfying $f(0) \leq 1$. Then*

$$\overline{\lim_{\epsilon \to +\infty}} \, G_f(\epsilon) = \sqrt{\overline{\lim_{\epsilon \to +\infty}} \frac{\epsilon^2}{-2 \log f(\epsilon)}}.$$

Theorems 4.3 and 4.4 give a useful criterion characterizing GDP and deepen our understanding of GDP. Putting the exact value of $\mu$ aside, a GDP algorithm must provide an infinite union of $(\epsilon, \delta)$-DP conditions, where $\delta$ must be $O(e^{-\epsilon^2})$ as $\epsilon \to \infty$. Refer to Appendices A.3 and A.4 for proofs of Theorems 4.3 and 4.4, respectively.

Using these newly-developed tools, we revisit our previous three examples for which the limit in Theorem 4.4 is $0$, $\sqrt{1/2}$, and $+\infty$, respectively. From these evaluations, we can conclude that the Laplace mechanism and SGD are GDP for some $\mu$ and that the privacy profile of the ICEA algorithm crosses every $\mu$-GDP curve regardless of how large $\mu$ is, indicating that the ICEA algorithm is not GDP.

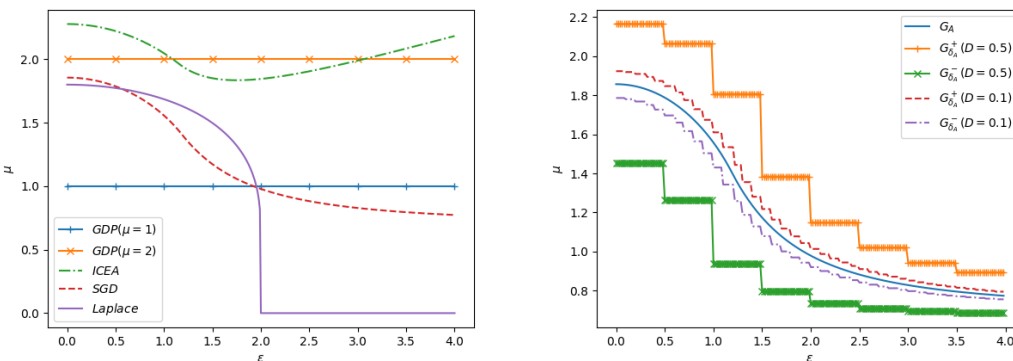

Figure 2: (Left) Examples of GDPTs. (Right) Plot of $G^+$ and $G^-$ with different values of $D$.

The left pane of Figure 2 shows the GDPTs of the three examples considered in this paper. All three GDPTs converge to a finite value as $\epsilon \to 0^+$. This can be attributed to the fact that any algorithm providing some non-trivial $(\epsilon, \delta)$-DP guarantee is $(0, \delta)$-DP for some $\delta \in [0, 1)$. For larger values of $\epsilon$, the GDPT of the Laplace mechanism takes on a constant value of $0$, the GDPT of SGD converges to a value that is approximately $0.7$, and the GDPT of the ICEA seems to diverging. These observations are consistent with the values of $0$, $\sqrt{1/2}$, and $\infty$ obtained from Theorem 4.4. Nonetheless, plots are only good for visualization and are not sufficient proof when verifying GDP. We still need objective and tractable methods for obtaining bounds on GDPTs.

Once an algorithm is confirmed to be GDP via Theorems 4.3 and 4.4, it is natural to be interested in the exact level of privacy protection, quantified by $\mu$. Following the intuition outlined by definition 3.1 and 3.2, we decompose the GDP condition into head and tail conditions and first focus on finding $\mu$ such that $\mathcal{A}$ is $(\epsilon, \mu)$-head GDP.

### 4.1 MEASURING THE HEAD

Without additional knowledge, finding $\sup\{G_{\delta_\mathcal{A}}(\epsilon) \mid \epsilon \in [0, \epsilon_h]\}$, even for a finite $\epsilon_h$, seems computationally infeasible as there are uncountably many real numbers in any non-empty interval and it is impossible to iterate through an infinite set. To solve this problem, we take advantage of the fact that $\mu_{\text{GDP}}$ has a bounded partial derivative.

**Theorem 4.5** *One of the partial derivatives of $\mu_{\text{GDP}}$ is uniformly bounded. Specifically, $0 \leq \frac{\partial \mu_{\text{GDP}}(\epsilon, \delta)}{\partial \epsilon} \leq \frac{\sqrt{2}\pi}{2}$.*

**Theorem 4.6** *Given $\epsilon_h \geq 0$, let $D = \epsilon_h/n$ and $x_i = iD + \epsilon_h$ for $i \in \{0, \ldots, n+1\}$. The GDPT of $\mathcal{A}$, denoted by $G_\mathcal{A}(\epsilon)$, is bounded between the two staircase functions*

$$G_{\delta_\mathcal{A}}^-(\epsilon) = \sum_{i=0}^{n+1} \mu_{\text{GDP}}(x_i, \delta_\mathcal{A}(x_{i+1})) \times \mathbf{1}_{\epsilon \in [x_i, x_{i+1})}$$

*and*

$$G_{\delta_\mathcal{A}}^+(\epsilon) = \sum_{i=0}^{n+1} \mu_{\text{GDP}}(x_{i+1}, \delta_\mathcal{A}(x_i)) \times \mathbf{1}_{\epsilon \in [x_i, x_{i+1})}.$$

*Specifically,*

$$\max_{i \in \{0,\ldots,n\}} G_{\delta_\mathcal{A}}^-(x_i) \leq \max_{\epsilon \in [0,\epsilon_h]} G_\mathcal{A}(\epsilon) \leq \max_{i \in \{0,\ldots,n+1\}} G_{\delta_\mathcal{A}}^+(x_i) \leq \max_{i \in \{0,\ldots,n\}} G_{\delta_\mathcal{A}}^-(x_i) + \sqrt{2}\pi D. \quad (3)$$

Refer to Appendix A.5 and A.6 for proofs of Theorem 4.5 and 4.6, respectively.

For any $\epsilon_h < +\infty$, we can now bound any GDPT $G_{\delta_{\mathcal{A}}}$ to any precision on $[0, \epsilon_h]$ without full point-wise evaluation because $G_{\delta_{\mathcal{A}}}$ is bounded between $G_{\delta_{\mathcal{A}}}^+$ and $G_{\delta_{\mathcal{A}}}^-$ and each staircase function takes on only finitely many values. The inequalities in (3) provide a viable way to estimate $\max G_{\delta_{\mathcal{A}}}(\epsilon)$: $\max_{i \in \{0,\ldots,n\}} G_{\delta_{\mathcal{A}}}^-(x_i)$ and $\max_{i \in \{0,\ldots,n+1\}} G_{\delta_{\mathcal{A}}}^+(x_i)$ can be computed by choosing the maximum value among $2n + 2$ evaluations of $\mu_{\text{GDP}}$. For the completeness, we provide a detailed description of an efficient algorithm with a time complexity of $O(\epsilon_h/D + \log^2(D) \log(\epsilon_h))$ in Appendix D.

## 4.2 CUTTING THE TAIL

With Theorem 4.6, one can verify $(\epsilon_h, \mu)$-head GDP conditions for arbitrarily large $\epsilon_h$ and an arbitrarily precise approximation of $\mu$. While the error in $\mu$ can be quantified by $D$, one gap remains: $\epsilon_h$ can be arbitrarily large but can never truly be $+\infty$. In this subsection, we discuss the gap between $(\epsilon_h, \mu)$-head GDP and true GDP (which is equivalent to $(+\infty, \mu)$-head GDP). Before giving a solution, we intuitively illustrate the gap between $(\epsilon_h, \mu)$-head GDP and true GDP. Consider the following two cases:

- GDP with catastrophic failure, where with probability $1 - p$, $\mathcal{A}_1$ functions properly and exact $\mu$-GDP is guaranteed, but with probability $p$, $\mathcal{A}_1$ malfunctions and discloses the entire dataset; and

- head-GDP with $\epsilon$-DP, where $\mathcal{A}_2$ is both $(\epsilon_h, \mu)$-head GDP and $(\epsilon_h, 0)$-DP.

The $\mu$-GDP privacy guarantee lies strictly between those of $\mathcal{A}_1$ and $\mathcal{A}_2$: specifically, $\delta_{\mathcal{A}_1}(\epsilon) < \delta_\mu(\epsilon) < \delta_{\mathcal{A}_2}(\epsilon)$.

In practice, $\mu$ is rarely above six in GDP and $\epsilon$ is rarely above 10 in $\epsilon$-DP because more-extreme values provide almost no privacy protection (Dong et al., 2019). If we verify the head condition up to $\epsilon_h = 100$ (which is not difficult because the time required for verification grows linearly) and take $\mu = 6$, then $p = \delta_\mu(\epsilon_h)$ will be on the order of $10^{-43}$. If $\epsilon_h$ is increased to 200, then $p$ will be smaller than $10^{-202}$. Hence, we conclude that the gap won't make any notable difference in practice with a proper choice of $\mu$ and $\epsilon_h$.

However, in some cases, one may still wish to fully mend the gap theoretically. This can also be achieved under the GDPT framework. A GDPT is a fixed function that inherits smoothness properties from the corresponding privacy profile. The maximum value of a GDPT can be analysed using standard functional tools. We give an example of such an analysis in Section 5. If the privacy profile is not given in a manageable closed form or the analysis is infeasible, we provide the following "clip and retify" procedure that can turn any $(\epsilon_h, \mu)$-head GDP algorithm into a $\mu$-GDP one at the cost of some utility.

**Theorem 4.7** *Let $\mathcal{A}$ be a $(\epsilon_h, \mu)$-head GDP algorithm with a numerical output. Assume that $-\infty < y^- < y^+ < +\infty$. Define $\mathcal{C}(y) = \max(\min(y, y^+), y^-)$ and $\mathcal{R}(z) = z + v$, where $v$ is sampled from $\text{Laplace}(b)$ with $b = (y^+ - y^-)/\epsilon_h$. Then $\mathcal{R} \circ \mathcal{C} \circ \mathcal{A}$ is $\mu$-GDP.*

Refer to Appendix A.7 for proof of Theorem 4.7. We remark that, in order to minimize the utility loss, the bounds $y^-$ and $y^+$ should be properly chosen and the head condition should be verified to an $\epsilon_h$ that is as large as possible.

# 5 APPLICATIONS

## 5.1 GDPT OF $\epsilon$-DP ALGORITHMS AND THE LAPLACE MECHANISM

By our previous analysis of the GDPT, we know that being GDP means that a privacy profile has a quickly vanishing tail (i.e., $\delta(\epsilon)$ must be $O(e^{-\epsilon^2})$). It is remarkable that another single parameter family of DP conditions, the $\epsilon$-DP conditions, is also a property that pertains to the tail of privacy profiles. For any $\epsilon_0$-DP algorithm, the privacy profile must be exactly $0$ after $\epsilon_0$. This suggests that $\epsilon_0$-DP is stronger than GDP. Next, we will quantify this intuition using the tools developed in this paper.

By Theorem 3.2, we know if $\mathcal{A}$ is $\epsilon_0$-DP, then in the worst case, $\delta_{\mathcal{A}}(\epsilon) = (e^{\epsilon_0} - e^{\epsilon})^+/(1 + e^{\epsilon_0})$.

We consider the GDPT of $\delta_{\mathcal{A}}$, denoted by $G_{\delta_{\mathcal{A}}}$. It is easy to see that, for $\epsilon \geq \epsilon_0$, $G_{\delta_{\mathcal{A}}}(\epsilon) = 0$: we need only consider $\epsilon \in [0, \epsilon_0)$. Let $G_{\delta_{\mathcal{A}}(\epsilon)}$ be denoted by $\mu_\epsilon$. Using using the partial derivative of $G_{\delta_{\mathcal{A}}}$ derived in Appendix A.5, we know that $\frac{\partial}{\partial \epsilon} G_{\delta_{\mathcal{A}}(\epsilon)} = \sqrt{2\pi} \exp\left\{ \left(\mu_\epsilon^2 + 2\epsilon\right)^2 /(8\mu_\epsilon^2) \right\} \left[ \Phi(-\frac{\mu_\epsilon^2 + 2\epsilon}{2\mu_\epsilon}) - \Phi(\frac{-\mu_0}{2}) \right]$. Then $\text{sign}(\frac{\partial}{\partial \epsilon} G_{\delta_{\mathcal{A}}(\epsilon)}) = \text{sign}(\mu_\epsilon - \mu_0 - 2\epsilon/\mu_0)$. We can conclude that $\mu_\epsilon \leq \mu_0$ and, further, that $G_{\delta_{\mathcal{A}}}(\epsilon)$ is strictly decreasing on $[0, \epsilon_0)$. By Theorem 4.1, we know that $\mathcal{A}$ is $\mu_0$-DP. This finding can be more generally formulated as the following theorem.

**Theorem 5.1** *Any $(\epsilon, 0)$-DP algorithm is also $\mu$-GDP for $\mu \geq -2\Phi^{-1}(1/(1 + e^{\epsilon}))$.*

Dong et al. (2019) pointed out that the DP guarantees of the Laplace mechanism are stronger than those correspondingly provided by $\epsilon$-DP. We reaffirm this difference by showing that it still exists under the GDP framework. The Laplace mechanism satisfies $\mu$-GDP for $\mu$ smaller than the bound given in Theorem 5.1. The GDPTs presented in Appendix E.1 illustrate this difference.

## 5.2 Utility refinement

There is an interesting byproduct or the privacy profile refinement. Theoretically, the privacy profile refinement can also be used to improve an algorithm's utility. For example, the projected noisy SGD algorithm in Feldman et al. (2018) is $(\epsilon, \delta)$-DP and the trade-off function is $\sigma = -C \log(\delta_0)/\epsilon_0$. To achieve $(0.2, e^{-2})$-DP, it appears that $\sigma$ needs to be chosen as $-C \log(e^{-2})/0.2 = 10C$. By Corollary 3.1, $(\epsilon, \delta)$-DP implies $(0.2, e^{-2})$-DP when $\delta + (1 - \delta)(e^{\epsilon} - e^{0.2})^+/(1 + e^{\epsilon}) = e^{-2}$. Numerical methods suggest that, by choosing $\epsilon \approx 0.334$ and $\delta \approx 0.067$, $(\epsilon, \delta)$-DP implies $(0.2, e^{-2})$-DP but $\sigma = -C \log(\delta)/\epsilon \approx 8.086C < 10C$. Therefore, the desired level of DP can be achieved with a lower noise parameter. However, this type of refinement majorly affects privacy profile around the origin and therefore minor in practice. For the general case and details of this derivation, refer to Appendix C.

## 5.3 Effect of subsampling

It is well known that a mechanism can be improved from a privacy perspective by adding extra steps to perturb its input or output. One popular approach is the subsampling technique (Balle et al., 2020a). We omit the details of this method and use the following result from Balle et al. (2020a).

Let $\mathcal{A}$ be an $(\epsilon, \delta)$-DP algorithm and $\mathcal{S}_\gamma$ the Poisson subsampling procedure wherein each data point is retained with probability $\gamma$. Then $\mathcal{A} \circ \mathcal{S}_\gamma$ is $(\log(1 - \gamma + \gamma e^{\epsilon}), \gamma \delta)$-DP when two datasets are considered differ in one element if one dataset equals to the other dataset plus one additional element. Combining this result with the partial derivatives of $\mu_{\text{GDP}}$, the Poisson subsampling procedure can significantly decrease the value of $\mu$ around $\epsilon = 0$ but has no effect on the GDPT's tail. Therefore, in the worst case, a subsampling procedure does not improve the value of $\mu$ for a GDP algorithm. For detailed plots, refer to Appendix E.2.

## 6 Conclusion and discussion

In this paper, we provided both an analytical understanding of and engineering tools for the GDP framework. Using the new notions we proposed, we gave solutions to two problems in GDP: identification and measurement. We further provided many applications of the proposed tools.

Our developments in this paper suggest numerous interesting directions for future work. The "clip and retify" procedure can be extended to provide more versatility to methods for privacy amplifications by utilizing more advanced composition and post-processing results. GDPT is generalizable to other parameterized DP notions such as RDP and cDP and enriches the DP literature with its tractability and visualizability.

**Ethics Statement.** Our work is majorly development of mathematical tools for the differential privacy framework. We cannot foresee any immediate real-world impact and believe those mathematical tools will contribute to the developments of privacy-aware technologies.

**Reproducibility Statement.** For all our theoretical results, we confirm that all necessary assumptions are properly addressed, and all proofs are provided unless trivial. No datasets and interactive virtual environments are involved in this paper. There are no experimental results in this work. Figures are only for demonstrative reasons as main results are supported by proofs.

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

## A    APPENDIX: PROOFS

**Proof A.1** *Proof of Theorem 3.1:*

*Sufficiency:*

*When $\epsilon \geq \epsilon_0$, the sufficiency is trivial as $\delta = \delta_0$.*

*When $\epsilon < \epsilon_0$, given that $\mathcal{A}$ is $(\epsilon_0, \delta_0)$-DP, by the definition, for any pair of datasets $S$ and $S'$ that differ in the record of a single individual and any event $E$,*

$$P[\mathcal{A}(S) \in E] \leq e^{\epsilon_0} P\left[\mathcal{A}\left(S'\right) \in E\right] + \delta_0.$$

*When $P\left[\mathcal{A}\left(S'\right) \in E\right] \leq \frac{1-\delta_0}{1+e^{\epsilon_0}} := c_0,$*

$$
\begin{aligned}
P[\mathcal{A}(S) \in E] &\leq e^{\epsilon_0} P\left[\mathcal{A}\left(S'\right) \in E\right] + \delta_0 \\
&\leq (e^{\epsilon_0} + e^{\epsilon} - e^{\epsilon})P\left[\mathcal{A}\left(S'\right) \in E\right] + \delta_0 + \delta - \delta \\
&\leq e^{\epsilon_0} P\left[\mathcal{A}\left(S'\right) \in E\right] + \delta + (e^{\epsilon_0} - e^{\epsilon})c_0 + \delta_0 - \delta \\
&\leq e^{\epsilon} P\left[\mathcal{A}\left(S'\right) \in E\right] + \delta + (e^{\epsilon_0} - e^{\epsilon})c_0 - \frac{(1-\delta_0)(e^{\epsilon_0} - e^{\epsilon})}{1 + e^{\epsilon_0}} \\
&\leq e^{\epsilon} P\left[\mathcal{A}\left(S'\right) \in E\right] + \delta.
\end{aligned}
$$

*When $c_0 \leq P\left[\mathcal{A}\left(S'\right) \in E\right] \leq 1,$*

$$
\begin{aligned}
P[\mathcal{A}(S) \in E] &= 1 - P[\mathcal{A}(S) \in E^c] \\
&\leq 1 - e^{-\epsilon_0}(P\left[\mathcal{A}\left(S'\right) \in E^c\right] - \delta_0) \\
&= 1 - e^{-\epsilon_0}(1 - P\left[\mathcal{A}\left(S'\right) \in E\right] - \delta_0) \\
&= 1 - e^{-\epsilon_0} + e^{-\epsilon_0} P\left[\mathcal{A}\left(S'\right) \in E\right] + e^{-\epsilon_0}\delta_0 \\
&= 1 - e^{-\epsilon_0} + e^{-\epsilon_0}\delta_0 + \delta - \delta + (e^{-\epsilon_0} + e^{\epsilon} - e^{\epsilon})P\left[\mathcal{A}\left(S'\right) \in E\right] \\
&= e^{\epsilon} P\left[\mathcal{A}\left(S'\right) \in E\right] + \delta + 1 - e^{-\epsilon_0} + e^{-\epsilon_0}\delta_0 - \delta + (e^{-\epsilon_0} - e^{\epsilon})P\left[\mathcal{A}\left(S'\right) \in E\right] \\
&\leq e^{\epsilon} P\left[\mathcal{A}\left(S'\right) \in E\right] + \delta + 1 - e^{-\epsilon_0} + e^{-\epsilon_0}\delta_0 - \delta + (e^{-\epsilon_0} - e^{\epsilon})c_0 \\
&= e^{\epsilon} P\left[\mathcal{A}\left(S'\right) \in E\right] + \delta + (1 - \delta_0)(\frac{e^{-\epsilon_0} - e^{\epsilon}}{1 + e^{\epsilon_0}} - e^{-\epsilon_0}) + 1 - \delta \\
&\leq e^{\epsilon} P\left[\mathcal{A}\left(S'\right) \in E\right] + \delta + (1 - \delta_0)(\frac{e^{-\epsilon_0} - e^{\epsilon}}{1 + e^{\epsilon_0}} - e^{-\epsilon_0} + 1 + \frac{e^{\epsilon} - e^{\epsilon_0}}{1 + e^{\epsilon_0}}) \\
&= e^{\epsilon} P\left[\mathcal{A}\left(S'\right) \in E\right] + \delta.
\end{aligned}
$$

*Necessity:*

*We prove the necessity by giving a specific $(\epsilon_0, \delta_0)$-DP algorithm $\mathcal{A}$ such that $\delta_{\mathcal{A}}(\epsilon)$ is exactly $\delta_0 + \frac{(1-\delta_0)(e^{\epsilon_0}-e^{\epsilon})^+}{1+e^{\epsilon_0}}$.*

*Define $\Omega_e = \{1, 2, 3, 4\}$ and $\Omega_S = \{0, 1\}$. Let $\epsilon \geq 0$, $0 \leq \delta_0 \leq 1$ and denote $\frac{e^{\epsilon_0}}{1+e^{\epsilon_0}}$ as $\alpha_0$. Let $\mathcal{A}$ be a randomized algorithm that take a single point from $\Omega_S$ and generate output as follows:*

$$
\begin{cases}
P(\mathcal{A}(S) = 1 \mid S = 0) = \delta_0, \\
P(\mathcal{A}(S) = 2 \mid S = 0) = 0, \\
P(\mathcal{A}(S) = 3 \mid S = 0) = (1 - \delta_0)\alpha_0, \\
P(\mathcal{A}(S) = 4 \mid S = 0) = (1 - \delta_0)(1 - \alpha_0),
\end{cases}
\qquad
\begin{cases}
P(\mathcal{A}(S) = 1 \mid S = 1) = 0, \\
P(\mathcal{A}(S) = 2 \mid S = 1) = \delta_0, \\
P(\mathcal{A}(S) = 3 \mid S = 1) = (1 - \delta_0)(1 - \alpha_0), \\
P(\mathcal{A}(S) = 4 \mid S = 1) = (1 - \delta_0)\alpha_0.
\end{cases}
$$

*By definition, $\delta(\epsilon)$ is the smallest $\delta$ such that $P(\mathcal{A}(S) \subset E \mid S = s) \leq e^{\epsilon} P(\mathcal{A}(S) \subset E \mid S = 1-s) + \delta$ holds true for all $E \subset \Omega_e$ and $s \in \Omega_S$. By checking all 64 combinations, we can conclude that $\delta_{\mathcal{A}}(\epsilon) = \delta_0 + \frac{(1-\delta_0)(e^{\epsilon_0}-e^{\epsilon})^+}{1+e^{\epsilon_0}}$.*

We present a key lemma underlying much of the theoretical analysis and may be of use for future developments.

**Lemma A.1** *Define $\tilde{\delta}_\mu(\epsilon) = \frac{\mu e^{-a^2/2}}{\sqrt{2\pi a^2}}$, where $a = -\frac{\epsilon}{\mu} + \frac{\mu}{2}$. It follows that $\lim_{\epsilon \to +\infty} \frac{\delta_\mu(\epsilon)}{\tilde{\delta}_\mu(\epsilon)} = 1$.*

**Proof A.2** *Proof of Lemma A.1:*

*It is well known that (Abramowitz et al., 1988), for $t < 0$:*

$$\frac{1}{-t + \sqrt{t^2 + 4}} < \sqrt{\frac{\pi}{2}} \exp\left(\frac{t^2}{2}\right) \Phi(t) < \frac{1}{-t + \sqrt{t^2 + \frac{8}{\pi}}}.$$

*Let $a = \left(-\frac{\varepsilon}{\mu} + \frac{\mu}{2}\right)$ and $b = \left(-\frac{\varepsilon}{\mu} - \frac{\mu}{2}\right)$,*

$$
\begin{aligned}
\overline{\lim_{\epsilon \to \infty}} \, \delta_\mu(\epsilon) &= \overline{\lim_{\epsilon \to \infty}} \, \Phi(a) - e^\epsilon \Phi(b) \\
&\leq \sqrt{\frac{2}{\pi}} \overline{\lim_{\epsilon \to \infty}} \frac{\exp\left(\frac{-a^2}{2}\right)}{-a + \sqrt{a^2 + \frac{8}{\pi}}} - \frac{\exp\left(\frac{-b^2}{2} + \epsilon\right)}{-b + \sqrt{b^2 + 4}}. \\
&= \sqrt{\frac{2}{\pi}} \overline{\lim_{\epsilon \to \infty}} \exp\left(\frac{-a^2}{2}\right) \left( \frac{1}{-a + \sqrt{a^2 + \frac{8}{\pi}}} - \frac{1}{-b + \sqrt{b^2 + 4}} \right). \\
&\leq \sqrt{\frac{2}{\pi}} \overline{\lim_{\epsilon \to \infty}} \exp\left(\frac{-a^2}{2}\right) \left(\frac{-1}{a}\right). \\
&= 0.
\end{aligned}
$$

$$
\begin{aligned}
\underline{\lim_{\epsilon \to \infty}} \, \delta_\mu(\epsilon) &= \underline{\lim_{\epsilon \to \infty}} \, \Phi(a) - e^\epsilon \Phi(b) \\
&\geq \sqrt{\frac{2}{\pi}} \underline{\lim_{\epsilon \to \infty}} \frac{\exp\left(\frac{-a^2}{2}\right)}{-a + \sqrt{a^2 + 4}} - \frac{\exp\left(\frac{-b^2}{2} + \epsilon\right)}{-b + \sqrt{b^2 + \frac{8}{\pi}}}. \\
&= \sqrt{\frac{2}{\pi}} \underline{\lim_{\epsilon \to \infty}} \exp\left(\frac{-a^2}{2}\right) \left( \frac{1}{-a + \sqrt{a^2 + 4}} - \frac{1}{-b + \sqrt{b^2 + \frac{8}{\pi}}} \right). \\
&\geq \sqrt{\frac{2}{\pi}} \underline{\lim_{\epsilon \to \infty}} \exp\left(\frac{-a^2}{2}\right) \left(\frac{-1}{b}\right). \\
&= 0.
\end{aligned}
$$

*Therefore,*

$$\lim_{\epsilon \to \infty} \delta_\mu(\epsilon) = 0. \tag{4}$$

*It is easy to see that,*

$$\lim_{\epsilon \to \infty} \tilde{\delta}_\mu(\epsilon) = \lim_{\epsilon \to \infty} \frac{\mu e^{-a^2/2}}{\sqrt{2\pi a^2}} = 0 \tag{5}$$

*By L'Hospital's rule:*

$$\lim_{\epsilon\to\infty} \frac{\tilde{\delta}_\mu(\epsilon)}{\delta_\mu(\epsilon)} = \lim_{\epsilon\to\infty} \frac{\tilde{\delta}'_\mu(\epsilon)}{\delta'_\mu(\epsilon)}$$

$$= \lim_{\epsilon\to\infty} -\frac{e^{-\frac{a^2}{2}}\left(a^2+2\right)}{\sqrt{2\pi}a^3} \Bigg/ e^\epsilon \Phi(b)$$

$$= \lim_{\epsilon\to\infty} \frac{e^{-\frac{b^2}{2}}\Phi(b)}{\sqrt{2\pi}b}$$

$$= \lim_{b\to-\infty} \frac{e^{-\frac{b^2}{2}}\Phi(b)}{\sqrt{2\pi}b}$$

$$= 1.$$

**Proof A.3** *Proof of Theorem 4.3:*

*It is easy to see that $\mathcal{A}$ is $(+\infty, +\infty)$-tail GDP if and only if $\overline{\lim}_{\epsilon\to+\infty} G_{\delta_\mathcal{A}}(\epsilon) < +\infty$.*

*Sufficiency:*

*If $\mathcal{A}$ is $\mu$-GDP. Then $\overline{\lim}_{\epsilon\to+\infty} G_{\delta_\mathcal{A}}(\epsilon) \leq \overline{\lim}_{\epsilon\to+\infty} G_{\delta_\mu}(\epsilon) = \mu$.*

*Necessity:*

*If $\overline{\lim}_{\epsilon\to+\infty} G_{\delta_\mathcal{A}}(\epsilon) = \mu < +\infty$, there must be a $\epsilon_t > 0$ such that $\mathcal{A}$ is $(\epsilon_t, \mu_0 + 1)$-tail GDP.*

*Notice that $\lim_{\mu\to\infty} \delta_\mu(\epsilon_t) = 1$, we can pick $\mu_1 > \mu_0$ large enough such that $\delta_{\mu_1}(\epsilon_t) > \delta_\mathcal{A}(0)$.*

*This is possible because by Theorem 3.1, $\delta_\mathcal{A}(0) < 1$. Then for $\epsilon \in [0, \epsilon_t)$, $\delta_\mathcal{A}(\epsilon) \leq \delta_\mathcal{A}(0) \leq \delta_{\mu_1}(\epsilon_t) \leq \delta_{\mu_1}(\epsilon)$. $\mathcal{A}$ is both $(\epsilon_t, \mu)$-head and tail GDP for $\mu = \mu_0 + \mu_1 + 1$. $\mathcal{A}$ is GDP as desired.*

**Proof A.4** *Proof of Theorem 4.4:*

*Let $\overline{\lim}_{\epsilon\to+\infty} G_f(\epsilon) = \mu_t$.*

*First we show that $\overline{\lim}_{\epsilon\to\infty} \frac{\epsilon^2}{-2\log\delta_\mathcal{A}(\epsilon)} \leq \mu_t^2$:*

*By the definition the limit, for any $\mu_0 > \mu_t$, for sufficient large $\epsilon$, $G_f(\epsilon) < \mu_0$ and further $\delta_\mathcal{A}(\epsilon) \leq \delta_{\mu_0}(\epsilon)$. Hence, $\overline{\lim}_{\epsilon\to\infty} \frac{\delta_\mathcal{A}(\epsilon)}{\delta_{\mu_0}(\epsilon)} \leq 1$. By Lemma A.1, $\overline{\lim}_{\epsilon\to\infty} \frac{\delta_\mathcal{A}(\epsilon)}{\tilde{\delta}_{\mu_0}(\epsilon)} \leq 1$.*

*Then $\lim_{\epsilon\to\infty} \frac{\epsilon^2}{-2\log\delta_\mathcal{A}(\epsilon)} \leq \lim_{\epsilon\to\infty} \frac{\epsilon^2}{-2\log\tilde{\delta}_{\mu_0}(\epsilon)} = \mu_0^2$.*

*$\lim_{\epsilon\to\infty} \frac{\epsilon^2}{-2\log\delta_\mathcal{A}(\epsilon)} \leq \mu_t$ as desired as we take $\mu_0 \to \mu_t$.*

*Next we show that $\overline{\lim}_{\epsilon\to\infty} \frac{\epsilon^2}{-2\log\delta_\mathcal{A}(\epsilon)} \geq \mu_t^2$:*

*If $\overline{\lim}_{\epsilon\to\infty} \frac{\epsilon^2}{-2\log\delta_\mathcal{A}(\epsilon)} = \mu_0^2 < \mu_t^2$, then by Lemma A.1,*

$$\overline{\lim}_{\epsilon\to\infty} \frac{\epsilon^2}{-2\log\delta_\mathcal{A}(\epsilon)} - \frac{\epsilon^2}{-2\log\delta_{\mu_t}(\epsilon)} = \overline{\lim}_{\epsilon\to\infty} \frac{\epsilon^2}{-2\log\delta_\mathcal{A}(\epsilon)} - \overline{\lim}_{\epsilon\to\infty} \frac{\epsilon^2}{-2\log\tilde{\delta}_{\mu_t}(\epsilon)}$$
$$< \mu_0^2 - \mu_t^2$$

*Then for a sufficiently large $\epsilon_0$,*

$$\frac{\epsilon_0^2}{-2\log\delta_\mathcal{A}(\epsilon_0)} - \frac{\epsilon_0^2}{-2\log\delta_{\mu_0}(\epsilon_0)} < 0.$$

*Since $\log$ is an increasing function, it follows that $\delta_\mathcal{A}(\epsilon_0) < \delta_{\mu_0}(\epsilon_0)$. Then $\overline{\lim}_{\epsilon\to+\infty} G_f(\epsilon) \leq \mu_0 < \mu_t$, which is a contradiction.*

**Proof A.5** *Proof of Theorem 4.5:*

*Let $G_\mu(\epsilon) = F(\epsilon, \delta_\mu(\epsilon))$ and $F(x, y) = \mu_{GDP}(x, y)$.*

*By definition of $\mu_{GDP}$, $G_\mu(\epsilon) = \mu$.*

*On one hand,*
$$\begin{cases} \dfrac{\partial G_\mu(\epsilon)}{\partial \epsilon} = \dfrac{\partial \mu}{\partial \epsilon} = 0, \\ \dfrac{\partial G_\mu(\epsilon)}{\partial \mu} = \dfrac{\partial \mu}{\partial \mu} = 1. \end{cases}$$

*On the other hand, by chain rule,*
$$\begin{cases} \dfrac{\partial G_\mu(\epsilon)}{\partial \epsilon} = \dfrac{\partial F}{\partial x} + \dfrac{\partial F}{\partial y}\dfrac{\partial \delta_\mu(\epsilon)}{\partial \epsilon}, \\ \dfrac{\partial G_\mu(\epsilon)}{\partial \mu} = \dfrac{\partial F}{\partial y}\dfrac{\partial \delta_\mu(\epsilon)}{\partial \mu}. \end{cases}$$

*Therefore,*
$$\begin{cases} \dfrac{\partial F}{\partial y} = (\dfrac{\partial \delta_\mu(\epsilon)}{\partial \mu})^{-1}, \\ \dfrac{\partial F}{\partial x} = -(\dfrac{\partial \delta_\mu(\epsilon)}{\partial \mu})^{-1}\dfrac{\partial \delta_\mu(\epsilon)}{\partial \epsilon}. \end{cases}$$

*Using the close forms, $\frac{\partial \delta_\mu(\epsilon)}{\partial \epsilon}$ and $\frac{\partial \delta_\mu(\epsilon)}{\partial \mu}$ can be directly computed:*

$$\begin{cases} \dfrac{\partial \delta_\mu(\epsilon)}{\partial \epsilon} = -e^\epsilon \Phi(-\dfrac{\mu^2 + 2\epsilon}{2\mu}), \\ \dfrac{\partial \delta_\mu(\epsilon)}{\partial \mu} = \dfrac{e^{-\frac{\left(\mu^2 - 2\epsilon\right)^2}{8\mu^2}}}{\sqrt{2\pi}}. \end{cases}$$

*Hence,*
$$\begin{cases} \dfrac{\partial F}{\partial x} = \sqrt{2\pi}e^{\frac{\left(\mu^2 + 2\epsilon\right)^2}{8\mu^2}}\Phi(-\dfrac{\mu^2 + 2\epsilon}{2\mu}) \le \sqrt{2\pi}e^{\frac{\mu^2}{8}}\Phi(-\dfrac{\mu}{2}) \le \dfrac{\sqrt{2\pi}}{2}, \\ \dfrac{\partial F}{\partial y} = \sqrt{2\pi}e^{\frac{\left(\mu^2 - 2\epsilon\right)^2}{8\mu^2}} > 0. \end{cases}$$

*Notice that $\frac{\partial F}{\partial x} = \sqrt{2\pi}e^{\frac{\left(\mu^2 + 2\epsilon\right)^2}{8\mu^2}}\Phi(-\frac{\mu^2 + 2\epsilon}{2\mu}) > 0$, combined with the fact that $\frac{\partial F}{\partial x} \le \frac{\sqrt{2\pi}}{2}$, we can conclude that $0 \le \frac{\partial \mu_{GDP}(\epsilon, \delta)}{\partial \epsilon} \le \frac{\sqrt{2\pi}}{2}$. By $\frac{\partial F}{\partial y} > 0$, we can see GDPT is order preserving.*

**Proof A.6** *Proof of Theorem 4.6:*

*We now consider the gap between $\max_{i \in \{0, \cdots, n\}}\{G_{\delta_\mathcal{A}}^-(x_i)\}$ and $\max_{i \in \{0, \cdots, n+1\}}\{G_{\delta_\mathcal{A}}^+(x_i)\}$ bound the length of $[\mu^-, \mu^+]$ in two cases.*

*Case 1:* *If $\max_{i \in \{0, \cdots, n+1\}}\{G_{\delta_\mathcal{A}}^+(x_i)\} = G_{\delta_\mathcal{A}}^+(x_0)$, then $\max_{i \in \{0, \cdots, n+1\}}\{G_{\delta_\mathcal{A}}^+(x_i)\} = G_{\delta_\mathcal{A}}^+(x_0) = \mu_{GDP}(D, \delta_\mathcal{A}(0)) \le \mu_{GDP}(0, \delta_\mathcal{A}(0)) + \frac{\sqrt{2\pi}D}{2}$. Therefore,*

$$\max_{\epsilon \in [0, \epsilon_h]} G(\epsilon) \le G_{\delta_\mathcal{A}}^+(x_0) \le \{G_{\delta_\mathcal{A}}^-(x_0)\} + \frac{\sqrt{2\pi}D}{2}.$$

*Case 2:* *If $\max_{i \in \{0, \cdots, n+1\}}\{G_{\delta_\mathcal{A}}^+(x_i)\} \ne G_{\delta_\mathcal{A}}^+(x_0)$, then by the order preserving property, the optimal $\mu$ lies in $[\mu^-, \mu^+]$, where $\mu^- = \max(\mu_h, \max_{i \in \{0, \cdots, n\}}\{G_{\delta_\mathcal{A}}^-(x_i)\})$ and $\mu^+ = \max(\mu_h, \max_{i \in \{1, \cdots, n+1\}}\{G_{\delta_\mathcal{A}}^+(x_i)\})$. Notice that*

$$\begin{aligned} \max_{i \in \{0, \cdots, n\}}\{G_{\delta_\mathcal{A}}^-(x_i)\} &= \max_{i \in \{0, \cdots, n\}}\{\mu_{GDP}(x_i, \delta_\mathcal{A}(x_{i+1}))\} = \max_{i \in \{1, \cdots, n+1\}}\{\mu_{GDP}(x_{i-1}, \delta_\mathcal{A}(x_i))\} \\ &\ge \max_{i \in \{1, \cdots, n+1\}}\{\mu_{GDP}(x_{i+1}, \delta_\mathcal{A}(x_i)) - \sqrt{2\pi}D\} \\ &\ge \max_{i \in \{1, \cdots, n+1\}}\{G_{\delta_\mathcal{A}}^+(x_i)\} - \sqrt{2\pi}D. \end{aligned}$$

*In both cases the gap is no greater than $\sqrt{2\pi}D$ as desired.*

**Proof A.7** *Proof of Theorem 4.7:*

*By the definition of $\mathcal{C}$, $\mathcal{C} \circ \mathcal{A}$ is bounded in $[y^-, y+]$. Therefore the global sensitivity of $\mathcal{C} \circ \mathcal{A}$ is no greater than $y + -y^-$. Then $\mathcal{R} \circ \mathcal{C} \circ \mathcal{A}$ is a special case of the Laplace mechanism. By Balle et al. (2020a), $\mathcal{R} \circ \mathcal{C} \circ \mathcal{A}$ is $\epsilon_h$-DP. Then $\delta_{\mathcal{R} \circ \mathcal{C} \circ \mathcal{A}}(\epsilon) = 0 < \delta_\mu(\epsilon)$ for any $\epsilon \geq \epsilon_h$.*

*In addition, because of the post-processing property, $\delta_{\mathcal{R} \circ \mathcal{C} \circ \mathcal{A}}(\epsilon) \leq \delta_{\mathcal{A}}(\epsilon) < \delta_\mu(\epsilon)$ for any $\epsilon < \epsilon_h$.*

*Therefore, $\mathcal{R} \circ \mathcal{C} \circ \mathcal{A}$ is $\mu$-GDP.*

# B APPENDIX: REFINING THE PRIVACY PROFILE

Given a trade-off function $\sigma = f(\epsilon, \delta)$ and a fixed parameter $\sigma$. From definition of the trade-off function it is instant that the for any $(\epsilon, \delta) \in \Omega = \{(\epsilon, \delta) \mid \sigma = f(\epsilon, \delta)\}$, $(\epsilon, \delta)$-DP is guaranteed. Then, $(\epsilon, \delta)$-DP is also guaranteed if there is a $(\epsilon_0, \delta_0) \in \Omega$ such that $(\epsilon_0, \delta_0)$-DP implies $(\epsilon, \delta)$-DP. Therefore,

$$\delta_{\mathcal{A}}(\epsilon) = \min\left(\{\delta \mid \sigma = f(\epsilon_0, \delta_0) \text{ and } \delta \geq \delta_0 + \frac{(1-\delta_0)(e^{\epsilon_0} - e^\epsilon)^+}{1 + e^{\epsilon_0}}\}\right).$$

Notice that by Corollary 3.2, $(\epsilon_0, \delta_0)$-DP implies $(\epsilon, \delta)$ with $\delta < \delta_0$ only if $\epsilon < \epsilon_0$, we rewrite the $\delta_{\mathcal{A}}(\epsilon)$ as:

$$\delta_{\mathcal{A}}(\epsilon) = \inf_{\epsilon_0 \in [\epsilon, \infty)} g(\epsilon, \epsilon_0),$$

where $g(\epsilon, \epsilon_0) := (1 - \hat{\delta}_{\mathcal{A}}(\epsilon_0))\frac{e^{\epsilon_0} - e^\epsilon}{e^{\epsilon_0} + 1} + \hat{\delta}_{\mathcal{A}}(\epsilon_0)$ and $\hat{\delta}_{\mathcal{A}}$ is the naive privacy profile defined implicitly by $\sigma = f(\epsilon_0, \delta_0)$. For continuously differentiable $f$, the minimum value of the right-hand side can be found be take the derivative:

$$\frac{\partial g(\epsilon, \epsilon_0)}{\partial \epsilon_0} = \frac{1 + e^\epsilon}{(1 + e^{\epsilon_0})^2}\left[\hat{\delta}_{\mathcal{A}}'(\epsilon_0) + e^{\epsilon_0}(1 - \hat{\delta}_{\mathcal{A}}(\epsilon_0) + \hat{\delta}_{\mathcal{A}}'(\epsilon_0))\right].$$

We remark that the sign of $\frac{\partial \delta_{\mathcal{A}}(\epsilon)}{\partial \epsilon_0}$ does not depend on $\epsilon$. For both of our example 2 and 3, we both find a particular value $\epsilon^i$ such that $Sign(\frac{\partial g(\epsilon, \epsilon_0)}{\partial \epsilon_0}) = -Sign(\epsilon - \epsilon^i)$. This means for $\epsilon \geq \epsilon^i$, $\delta_{\mathcal{A}}(\epsilon) = \hat{\delta}_{\mathcal{A}}(\epsilon)$ and otherwise $\delta_{\mathcal{A}}(\epsilon)$ equals to the $\delta$ value derived from $(\epsilon, \hat{\delta}_{\mathcal{A}}(\epsilon))$.

# C APPENDIX: REFINING THE $\sigma$

Given a trade-off function $\sigma = f(\epsilon, \delta)$ and desired $(\epsilon, \delta)$. From definition of the trade-off function, $(\epsilon, \delta)$-DP is guaranteed when $\sigma = f(\epsilon, \delta)$. We treat the $\sigma$ as the noise parameter and without loss of generality we assume a smaller value of $\sigma$ is preferred. Notice that $(\epsilon, \delta)$-DP is also guaranteed if there is a $(\epsilon_0, \delta_0) \in \Omega$ such that $(\epsilon_0, \delta_0)$-DP implies $(\epsilon, \delta)$-DP. Therefore, $\sigma$ can be chosen as,

$$\sigma^- = \min\left(\{f(\epsilon_0, \delta_0) \mid \delta \geq \delta_0 + \frac{(1-\delta_0)(e^{\epsilon_0} - e^\epsilon)^+}{1 + e^{\epsilon_0}}\}\right).$$

Notice that by Corollary 3.2, $(\epsilon_0, \delta_0)$-DP implies $(\epsilon, \delta)$ with $\delta < \delta_0$ only if $\epsilon < \epsilon_0$, we rewrite the $\sigma^-$ as:

$$\sigma^- = \inf_{\epsilon_0 \in [\epsilon, \infty)} g(\epsilon, \epsilon_0),$$

where $g(\epsilon, \epsilon_0) := f(\epsilon_0, (1 - \hat{\delta}_{\mathcal{A}}(\epsilon_0))\frac{e^{\epsilon_0} - e^\epsilon}{e^{\epsilon_0} + 1} + \hat{\delta}_{\mathcal{A}}(\epsilon_0))$ and $\hat{\delta}_{\mathcal{A}}$ is the naive privacy profile defined implicitly by $\sigma = f(\epsilon_0, \delta_0)$. For continuously differentiable $f$, the minimum value of the right-hand side can be found be take the derivative:

$$\frac{\partial g(\epsilon, \epsilon_0)}{\partial \epsilon_0} = \frac{1 + e^\epsilon}{(1 + e^{\epsilon_0})^2}\left[\hat{\delta}_{\mathcal{A}}'(\epsilon_0) + e^{\epsilon_0}(1 - \hat{\delta}_{\mathcal{A}}(\epsilon_0) + \hat{\delta}_{\mathcal{A}}'(\epsilon_0))\right]\frac{\partial f}{\partial \delta}\big|_{\epsilon=\epsilon_0, \delta=\delta_0} + \frac{\partial f}{\partial \epsilon}\big|_{\epsilon=\epsilon_0, \delta=\delta_0}.$$

We remark that the sign of $\frac{\partial \delta_{\mathcal{A}}(\epsilon)}{\partial \epsilon_0}$ does not depend on $\epsilon$ and the zero point can be effectively solved numerically.

## D  A SELF-CONTAINED EFFICIENT HEAD MEASUREMENT ALGORITHM

First we formalise the binary search algorithm to find $\mu_{\text{GDP}}$:

---
**Algorithm 1:** Binary search
---
Input: $\epsilon, \delta, \mu_{\max}, b$ ;
Output: $\mu_-, \mu_+$ (lower and upper bound of $\mu$).;
$\mu_- \leftarrow 0$;
$\mu_+ \leftarrow \mu_{\max}$;
**while** $\mu^+ - \mu^- > b$ **do**
 $\mu = \frac{\mu^+ + \mu^-}{2}$;
 **if** $\delta_\mu(\epsilon) > \delta$ **then**
  $\mu^+ \leftarrow \mu_0$;
 **else**
  $\mu^- \leftarrow \mu_0$;
 **end**

**end**
return $\mu^+, \mu^-$.

---

Given a proper searching range $[0, \mu_{\max}]$, the algorithm 1 yields $\mu^+$ and $\mu^-$ such that $\mu^- \leq \mu_{\text{GDP}}(\epsilon, \delta) \leq \mu^+$ and $\mu^+ - \mu^- < b$. We denote $\mu^+$ as $\mu_{\text{GDP}}^+(\epsilon, \delta, \mu_{\max}, b)$ and $\mu^-$ as $\mu_{\text{GDP}}^-(\epsilon, \delta, \mu_{\max}, b)$. When searching for $\mu$ via privacy profiles, the $\mu_{\max}$ can chosen as $\mu_{\text{GDP}}(\mu_t, \delta_A(0))$ because $\mu_{\text{GDP}}$ will not exceed this value on $[0, \epsilon_t]$ (refer to Theorem 4.3 for the value of $\mu_t$). Alternatively, $\mu_{\max}$ can be set to a large constant for convenience, for example 10. In this case, if the outputted $\mu^+$ equals to the preset value (10) then the privacy profile fails to imply 10-GDP. In practice, GDP with $\mu \geq 6$ provides almost no privacy protection Dong et al. (2019).

With the formal definition of binary search we use, the exhaustive iteration method to bound the staircase functions outlined in Theorem 4.6 can be formally written as follows:

---
**Algorithm 2:** Finding $\mu$ with privacy profiles (Naive).
---
Input: $\delta_{\mathcal{A}}, \epsilon_h, \mu_t, c, \mu_{\max}$. (Privacy profile, the end point of searching $\epsilon_h$, $\mu$ given by Theorem 4.3, reciprocal of error margin, searching range);
Output: $\mu_-, \mu_+$ (lower and upper bound of $\mu$).;
$i \leftarrow 0$;
$n \leftarrow \lceil \sqrt{8}c\pi\epsilon_h \rceil + 1$;
$D \leftarrow \frac{\epsilon_h}{n-1}$;
$\mu_- \leftarrow \max(\mu_h, \mu_t)$;
$\mu_+ \leftarrow \mu_{\max}$;
**while** $i \leq n + 1$ **do**
 $x^- \leftarrow iD + \epsilon_h$;
 $x^+ \leftarrow (i+1)D + \epsilon_h$;
 $\mu_+ \leftarrow \max(\mu_+, \mu_{\text{GDP}}^+(x^-, \delta_{\mathcal{A}}(x^+), \mu_{\max}, \frac{1}{2c}))$;
 $\mu_- \leftarrow \max(\mu_-, \mu_{\text{GDP}}^-(x^+), \delta_{\mathcal{A}}(x^-, \mu_{\max}, \frac{1}{2c}))$;
 $i \leftarrow i + 1$
**end**
return $\mu^+, \mu^-$.

---

Algorithm 2 naively go thorough all $G_{\delta_{\mathcal{A}}}^-(x_i)$ and $G_{\delta_{\mathcal{A}}}^+(x_i)$. By the choice of $D$, the true gap between $\max G_{\delta_{\mathcal{A}}}^-(\epsilon)$ and $\max G_{\delta_{\mathcal{A}}}^+(\epsilon)$ is less than $\frac{1}{2c}$ and the binary search estimate the $\mu_{\text{GDP}}$ with an error less than $\frac{1}{2c}$. Combined together, the overall gap between estimated $\mu^+$ and $\mu^-$ is less than $\frac{1}{c}$. However, there is still room for optimization:

We take $\mu_+ \leftarrow \max(\mu_+, \mu_{\text{GDP}}^+(x^-, \delta_{\mathcal{A}}(x^+), \mu_{\max}, \frac{1}{2c}))$ for example, same optimization can be applied to $\mu_- \leftarrow \max(\mu_-, \mu_{\text{GDP}}^-(x^+, \delta_{\mathcal{A}}(x^-), \mu_{\max}, \frac{1}{2c})))$ as well. The naive operation, $\mu_+ \leftarrow \max(\mu_+, \mu_{\text{GDP}}^+(x^-, \delta_{\mathcal{A}}(x^+), \mu_{\max}, \frac{1}{2c}))$ can be optimized into "If $\delta_{\mu^+}(x^-) < \delta_{\mathcal{A}}(x^+)$, then $\mu^+ \leftarrow \mu_{\text{GDP}}^+(x^-, \delta_{\mathcal{A}}(x^+), \mu_{\max}, \frac{1}{2c}))$. To see this, we list all three possibilities as follows:

- Case 1: $\mu^+ < \mu_{\text{GDP}}(x^-, \delta_{\mathcal{A}}(x^+)) \leq \mu_{\text{GDP}}^+(x^-, \delta_{\mathcal{A}}(x^+), \mu_{\max}, \frac{1}{2c}))$.
- Case 2: $\mu_{\text{GDP}}(x^-, \delta_{\mathcal{A}}(x^+)) \leq \mu^+ \leq \mu_{\text{GDP}}^+(x^-, \delta_{\mathcal{A}}(x^+), \mu_{\max}, \frac{1}{2c}))$.
- Case 3: $\mu_{\text{GDP}}(x^-, \delta_{\mathcal{A}}(x^+)) \leq \mu_{\text{GDP}}^+(x^-, \delta_{\mathcal{A}}(x^+), \mu_{\max}, \frac{1}{2c})) < \mu^+$.

In case 1, both of the naive operation and the optimized operation will update $\mu^+$ to $\mu_{\text{GDP}}^+(x^-, \delta_{\mathcal{A}}(x^+), \mu_{\max}, \frac{1}{2c}))$.

In case 2, the optimized operation will do nothing, because the test $\delta_{\mu^+}(x^-) < \delta_{\mathcal{A}}(x^+)$ will fail. The naive operation will update $\mu^+$ due to the error of binary search, which should be avoided.

In case 3, the optimized operation will do nothing, because the test $\delta_{\mu^+}(x^-) < \delta_{\mathcal{A}}(x^+)$ will fail. The naive operation will also do nothing because the max operator will choose $\mu^+$.

To sum up, the optimized operation always give a more accurate update.

Besides, we want to avoid case 1 because only in that case a binary search is needed. Notice that case 1 happens only if $\delta_{\mu^+}(x^-) < \delta_{\mathcal{A}}(x^+)$, which is equivalent to $\mu^+ < \mu_{\text{GDP}}(x^-, \delta_{\mathcal{A}}(x^+))$. In the $k+1$ round of loop, the condition $\mu^+ < \mu_{\text{GDP}}(x^-, \delta_{\mathcal{A}}(x^+))$ holds true only if for all $j \in \{0, \cdots, k\}$, $\mu_{\text{GDP}}(x_j^-, \delta_{\mathcal{A}}(x_j^+)) < \mu_{\text{GDP}}(x^-, \delta_{\mathcal{A}}(x^+))$, where $x_j^-$ and $x_j^+$ are the values of $x^-$ and $x^+$ in the round $j$. This inspire us to shuffle $x_i$ before iteration because after shuffling, the probability of "$\mu_{\text{GDP}}(x_j^-, \delta_{\mathcal{A}}(x_j^+)) < \mu_{\text{GDP}}(x^-, \delta_{\mathcal{A}}(x^+))$ for all $j \in \{0, \cdots, k\}$" will be $\frac{1}{k+1}$. The expected occurrence of case 1 will be $\sum_{k=0}^{n+1} \frac{1}{k+1} = O(\log(n))$. We present the complete optimized algorithm as follows:

---

**Algorithm 3:** Finding $\mu$ with privacy profiles (optimized).

---

Input: $\delta_{\mathcal{A}}, \epsilon_h, \mu_t, c, \mu_{\max}$. (Privacy profile, the end point of searching $\epsilon_h$, $\mu$ given by Theorem 4.3, reciprocal of error margin, searching range);
Output: $\mu_-, \mu_+$ (lower and upper bound of $\mu$).;
$i \leftarrow 0$;
$n \leftarrow \lceil \sqrt{8}c\pi\epsilon_h \rceil + 1$;
$D \leftarrow \frac{\epsilon_h}{n-1}$;
$\mu_- \leftarrow \max(\mu_h, \mu_t)$;
$\mu_+ \leftarrow \mu_{\max}$;
$\mathcal{S} = [0, 1, \cdots, n+1]$;
Shuffle $\mathcal{S}$;
**while** $i \leq n+1$ **do**
    $x^- \leftarrow S[i]D + \epsilon_h$;
    $x^+ \leftarrow S[i+1]D + \epsilon_h$;
    **if** $\delta_{\mu^+}(x^-) < \delta_{\mathcal{A}}(x^+)$ **then**
        |  $\mu^+ \leftarrow \mu_{\text{GDP}}^+(x^-, \delta_{\mathcal{A}}(x^+), \mu_{\max}, \frac{1}{2c}))$;
    **end**
    **if** $\delta_{\mu^-}(x^+) < \delta_{\mathcal{A}}(x^-)$ **then**
        |  $\mu^- \leftarrow \mu_{\text{GDP}}^-(x^+, \delta_{\mathcal{A}}(x^-), \mu_{\max}, \frac{1}{2c}))$;
    **end**
    $i \leftarrow i + 1$
**end**
return $\mu^+, \mu^-$.

---

The time complexity of shuffling $\mathcal{S}$ is $O(n) = O(\epsilon_h c)$. Each binary search has a time complexity of $O(\log(c))$ and the expected number of binary searches is $O(\log(\epsilon_h c))$. The overall time complexity of the optimized algorithm is therefore $O(\epsilon_h c + \log^2(c)\log(\epsilon_h))$.

# E   APPENDIX: PLOTS

## E.1   THE LAPLACE MECHANISM UNDER GDP

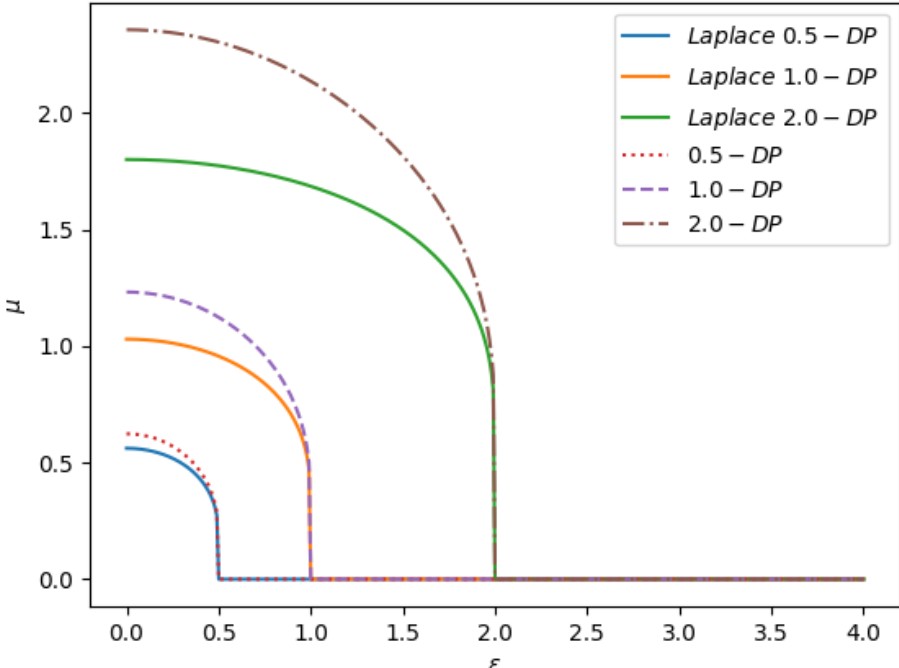

Figure 3: The plot of GDPT of $\epsilon$-DP privacy profiles and the Laplace mechanisms with the same $\epsilon$-DP guarantee.

## E.2 THE EFFECT OF SUBSAMPLING

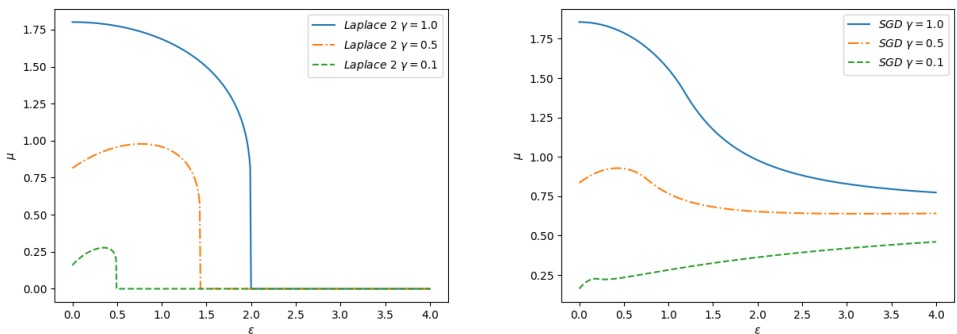

Figure 4: (Left) GDPT of the Laplace mechanism for various of $\gamma$. (Right) GDPT of the SGD for various of $\gamma$.

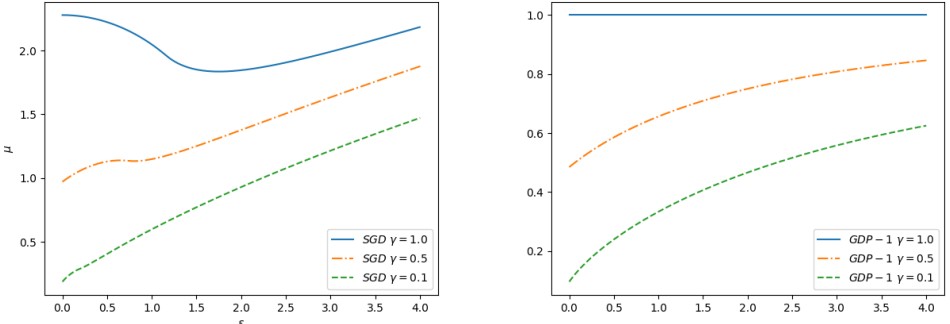

Figure 5: (Left) GDPT of the ICEA for various of $\gamma$. (Right) GDPT of the $\delta_\mu$ for various of $\gamma$.

