# OpenReview forum: "Gaussian Differential Privacy Transformation: from identification to application"
_ICLR.cc/2022/Conference — ICLR 2022 Submitted_

### Official Review · Reviewer_H1fA · 2021-10-30

**Correctness:** 4
**Technical Novelty And Significance:** 2
**Empirical Novelty And Significance:** 2
**Recommendation:** 5
**Confidence:** 3

**Main Review:**

## Strengths
- This paper proposes a transformation and several auxiliary lemmas that may help make GDP more practical and more widely applicable.
- The non-trivial relationship between $(\epsilon, \delta)$-DP's is interesting.

## Weaknesses
- I do not believe that GDPT yields significant insight that cannot be drawn from the privacy profile curve itself. Specifically, almost all the applications can be derived via the latter as well. (More details are presented below in "Detailed Comments for Authors".)
- Although the paper claims to provide "engineering tools" for GDP, no evaluation is given. On this front, I think it is important to show e.g. how tight & how efficient GDP tools presented here are compared to other methods in literature such as Privacy Loss Distributions [Sommer et al., PETS 2019].
- The "utility improvement" from the non-trivial relationship between $(\epsilon, \delta)$-DP's does not seem that appealing. Specifically, the examples in the paper uses very high $\delta$ (all of them larger than $0.05$) which is not a value used anywhere in practice. (I have not seen any practical application where $\delta > 10^{-4}$.) Indeed, this does not seem like a coincidence: the new relation only seem useful for $\epsilon, \epsilon_0$ relatively close and $\delta, \delta_0$ relatively large (otherwise the additive term in $\delta_0$ would be too large).
- The non-trivial relationship between $(\epsilon, \delta)$-DP's also seems to be implicit in previous work. As an example, the paper "Optimal Accounting of Differential Privacy via Characteristic Function" by Zhu et al. has a characterization that says that the privacy curve must be convex (Lemma 11) when the x-axis is changed from $\epsilon$ to $e^{\epsilon}$. Using this convexity on $0, \epsilon_0, \epsilon$ also yields this characterization. I think this is a minor weakness since it is good for the statement to be written down in a more explicit form as is done here anyway.

## Detailed Comments for Authors
- *Regarding GDPT:*
  - Theorem 4.4 can be easily derived from Lemma A.1 by setting up $f(\epsilon) \leq \delta_\mu(\epsilon)$ for some $\mu$.
  - For Theorem 4.6, the staircase functions taken are equivalent to rounding up or down $\delta(\epsilon)$ in the privacy profile for all $\epsilon \in [x_i, x_{i + 1}]$ in each $i$. So the entire proof of Theorem 4.6 can also be derived as easily via the privacy profile curve as well.
  - Overall, I think it'd be better if you show an application of GDPT that would be much harder to derive under the privacy profile notion. Otherwise, I'm not sure why we should define yet another privacy curve in addition to the already many curves.

### Other More Minor Comments:
- Definition 1.1: missing $\leq$ in the equation.
- Page 2, paragraph before Section 1.1: $\sigma$ is not yet defined?
- Page 3, second example: isn't the Gaussian mechanism itself also an example of such a tradeoff?
- Page 3, third example: the ICEA algorithm of Ghazi et al. (2019) was actually proposed much earlier by Ishai et al. [FOCS 2006] in the paper " Cryptography from anonymity". See also the papers "Private Aggregation from Fewer Anonymous Messages" by Ghazi et al. [EUROCRYPT 2020] and "Private Summation in the Multi-Message Shuffle Model" by Balle et al. [CCS 2021] which give better analysis of ICEA compared to Ghazi et al. (2019).
- Page 3, third example: m in the ICEA algorithm is *not* maximum error but the number of messages (i.e. additive shares) per user.
- Page 5 Definition 3.2: it doesn't seem like the tail condition is used in a meaningful way (at least in the main body)? If so, then maybe there is no need to define it (at least in the main body).
- Page 6 before Theorem 4.1: "... the following theorems *formalizes* ..." -> "... the following theorems *formalize* ..."

**Summary Of The Paper:**

## Summary of Contributions

This paper considers Gaussian differential privacy (GDP) notion which is an extension of $(\epsilon, \delta)$-differential privacy (DP). Roughly speaking, an algorithm is $\mu$-GDP if it is "at least as private as Gaussian mechanism with noise multiplier $1/\mu$". The main advantages of GDP are that (i) it has a simple and sharper composition theorem than $(\epsilon, \delta)$-DP and (ii) it has only a single parameter $\mu$ leading to simpler/more efficient computation compared to other DP notions that provide good composition (e.g. $f$-DP or Privacy Loss Distribution (PLD)).

This paper develops tools that can help understand/deploy GDP more easily. Specifically:
- They propose Gaussian Differential Privacy Transformation (GDPT) which roughly speaking is the curve $\mu: \mathbb{R} \to \mathbb{R}$ where $\mu(\epsilon)$ is the smallest $\epsilon$ for which the algorithm is at least as private as Gaussian mechanism of noise multiplier $1/\mu(\epsilon)$ at this particular value of $\epsilon$.
- With the above notion, they derive (in Theorems 4.3, 4.4) the condition for an algorithm to be $\mu$-GDP for some finite $\mu$: it must be $(\epsilon, \delta(\epsilon))$-DP where $\limsup \frac{\epsilon^2}{-\log \delta(\epsilon)}$ is finite.
- They then show how to approximately check whether an algorithm is GDP. Specifically, they show that it suffices to check a certain condition on a finite number of points (corresponding to an upper staircase in Figure 2 right). They show that if this condition holds the algorithm is $(\epsilon_h, \mu)$-head GDP, roughly meaning that it is "GDP" for all $\epsilon < \epsilon_h$.
- They argue that in practice taking $\epsilon_h$ to be sufficiently large should be enough. But they also give a "clip and noise" procedure to turn an $(\epsilon_h, \mu)$-head GDP algorithm into one which is actually $\mu$-GDP.


Along the way, the paper also shows a new implication between $(\epsilon, \delta)$-DP and $(\epsilon', \delta')$-DP. Previously, it was known that the former implies the latter if $\epsilon' \geq \epsilon$ and $\delta' \geq \delta$. In Theorem 3.1 of this paper, the authors show that the implication may even hold when $\epsilon > \epsilon'$ but $\delta'$ also has to be larger than $\delta$ by a certain amount (depending on $\delta, \epsilon, \epsilon'$). The authors give some example in Section 5 where this relation gives non-trivial implications for some known algorithms.

**Summary Of The Review:**

## Recommendation

Overall, although I think this paper advances the understanding of GDP, I am unconvinced that GDPT yields significant novel insight that cannot be achieved otherwise and more empirical evaluations have to be made in order to determine the practicality of the tools proposed here. Due to this, I recommend rejection.

---

> ### Author Response · Authors · 2021-11-22
> **Reply to reviewer H1fa**
>
>
> We thank the reviewer for the valuable remarks on our work. Below, we address all your concerns. Please let us know if there are any further questions.
>
> ## Why we should define yet another privacy curve? How is GDPT compared to other frameworks?
>
> There might be some misunderstanding. Unlike the RDP, the CDP or the PLD you mentioned, the role of GDPT is an intermediate tool instead of another privacy curve framework. Therefore, we didn't develop any composition theorem or design any new algorithm with a privacy guarantee given in the form of GDPT. Aside from providing more insights about GDP, the main goal of our paper is to build a tractable bridge between DP and GDP. The motivations behind this goal are the advantages of GDP itself and the rich literature and tools developed using the notation of DP. We treat the mechanisms as black boxes and only work on privacy guarantees that are known to us. One may argue that opening the black boxes and redesigning the mechanisms from scratch will lead to better outcomes. However, we want to point out that redesigning the mechanisms from scratch is not always possible, especially when the data has already been collected and perturbed.
>
> ## Theorem 4.6 can also be derived as easily via the privacy profile curve as well by rounding
>
> I'm afraid that it may not be as simple as it looks. The forms of staircase functions are carefully picked to make sure the gap between $G^+$ and $G^-$ can be controlled by the constant $D$ in a tractable way. The privacy profile $\delta(\epsilon)$ may decrease abruptly at some point, and the difference between upper and lower rounding may be significant.We want to avoid making any assumptions about the privacy profile $\delta(\epsilon)$, as these could be difficult to verify.Therefore, we turn to the properties of GDP itself and find that through the special form of staircase functions we pick, the density of sampling points can remain the same no matter how steep or flat the $\delta(\epsilon)$ is. Furthermore, by using the GDPT, the search for mu can be accomplished by analysing only one curve (the GDPT) rather than many curves with varying properties ($\delta_\mu$ for different $\mu$).
>
> ## Theorem 4.4 can be easily derived from Lemma A.1
>
> Theorem 4.4 is derived from Lemma A.1. Theorem 4.4 gives an easy-to-verify condition to the tail condition outlined in Theorem 4.3. GDP is a condition of the entire privacy curve, but by our analysis, the identification of GDP algorithms (without considering the exact value of $\mu$) is only related to the tail. We will clarify in Theorem 4.3.
>
> ## The "utility improvement" from the non-trivial relationship does not seem that appealing
>
> The overlooked partial order is theoretically critical. We agree that the improvement shown in the illustration is only an interesting side-effect of the theoretical development.  As is shown in Figure 1 (left), the corresponding refinement is significant for the privacy profile near the origin. Without the refinement, the trade-off function itself cannot correctly specify some algorithms (e.g., example 2 and 3) offers non-trivial $(0,\delta)$-DP guarantees and therefore cannot be correctly classified as GDP. We will clarify that the improvement is minor in 5.2.
>
> ## Using the convexity also yields this characterization of the non-trivial relationship
>
> As is shown by the necessity part in A.1, the non-trivial relationship we give is tight. Therefore, it is natural that any precise enough privacy profile manipulation should end up with the exact same conclusion.
>
> ## It'd be better if you show an application of GDPT that would be much harder to derive under the privacy profile notion.
>
> In a separate reply we will give an example to show what kind of difficulties we may encounter without the tools developed in our paper.
>
> ## Minor Comments
>
> •   Typo
>
> Fixed. Thank you for pointing out.
>
> •   Page 2: $\sigma$ is not yet defined?
>
> We explained in that sentence: $\sigma$ is utility parameter (usually the scale of noise).
>
> •   Second example: isn't the Gaussian mechanism itself also an example of such a tradeoff?
>
> It was. The tradeoff can now be tightly shown using the framework of GDP.
>
> •   The ICEA algorithm was actually proposed much earlier. See also ...
>
> We clarified that the ICEA was actually proposed much by Ishai et al. We added those two new references to that passage in case readers are interested in ICEA itself.
>
> •   Page 5 Definition 3.2: It doesn't seem like the tail condition is used in a meaningful way (at least in the main body)?
>
> As is stated in 4.2, the actual performance is most dominated by the head condition. However, the tail condition solely decides whether an algorithm is GDP or not. Before, it was only implicitly considered in Theorem 4.3. We changed the form of Theorem 4.3 as follows to clearly reflect the importance of the tail condition:
>
> (Theorem 4.3) An algorithm $\mathcal{A}$ is GDP if and only if $\mathcal{A}$ is $(+\infty,+\infty)$-tail GDP.

---

> > ### Author Response · Authors · 2021-11-22
> > **Example**
> >
> > # Example
> > In this section, we extracted Example 2 from our paper with additional comments on how our new tools fix what previous works are lacking.
> >
> > Theorem 3.2 (Theoretical conversion): A mechanism is $\mu$-GDP if and only if it is $(\epsilon, \delta_\mu(\epsilon))$-DP for all $\epsilon \geq 0$, where
> >
> >
> > $$\delta_\mu(\epsilon)=\Phi\left(-\frac{\epsilon}{\mu}+\frac{\mu}{2}\right)-\mathrm{e}^{\epsilon} \Phi\left(-\frac{\epsilon}{\mu}-\frac{\mu}{2}\right) .$$
> >
> >
> > In the example 2, suppose we are given an algorithm $X$ where the noise parameter has the form $\sigma=A\epsilon^{-1}\sqrt{\log (B / \delta)}$. For simplicity, we let $\sigma=A=B=1$.
> >
> > We start with the identification problem: is $X$ GDP?
> >
> >
> >
> >
> > ##Issues prior to our work
> > According to Theorem 3.2, $X$ is GDP if and only if the implicit $\delta(\epsilon)$ satisfies $\delta(\epsilon)\leq \delta_\mu(\epsilon)$ for some finite $\mu$. A naive application of Theorem 3.2 will lead to a failure: It is easy to see that $\delta_\mu(\epsilon)<1$ for any $\mu>0$ but $\delta(\epsilon)\geq 1$ for sufficiently small $\epsilon$. This naive comparison will disqualify $X$ as GDP by mistake.
> >
> > ## Our solution:
> >
> > To solve this problem, we give Theorem 3.1, Theorem 3.2 and Corollary 3.2 to show that $\delta(\epsilon)< 1$ as long as $X$ provides a single arbitrarily weak non-trival $(\epsilon,\delta)$-DP guarantee.
> >
> > ## Issues prior to our work:
> >
> >  Even with $\delta(\epsilon)$ correctly specified, the existence of $\mu$ such that $\delta(\epsilon)<\delta_\mu(\epsilon)$ for all $\epsilon\geq 0$ is not easy to decide.
> >
> > ## Our solution
> >
> > As we demonstrated in Theorem 4.3, the existence of $mu$ is solely determined by an asymptotic comparison of $\delta(\epsilon) $and $\delta_\mu(\epsilon) $at $+\infty$.The asymptotic characterization is given as Lemma A.1. Therefore, the solution to the identification problem is wrapped up in a simple theorem (Theorem 4.4).
> >
> > Then we move forward to the measurement problem. As a GDP algorithm, what is the $\mu$ of $X$?
> >
> > ## Issues prior to our work
> >  The Theorem 3.2 gives that $\mu$ is the smallest $\mu_0$ such that $\delta(\epsilon)\leq\delta_{\mu_0}(\epsilon)$ for all $\epsilon\geq 0$. We would be forced to search through an uncountably-large family of functions for a single $\delta_\mu(\epsilon)$ that never crosses $\delta(\epsilon)$ anywhere on $[0,\infty)$ and has $\mu$ as small as possible. Those conditions cannot be check numerically, even for a single $\mu$. Turning to closed forms for privacy profiles and $\delta_\mu$ is also difficult: even if a given privacy profile is easy to handle, $\delta_\mu$ presents some technical hurdles. The profile $\delta_\mu$ and $\Phi$ are transcendental with different asymptotic behaviors for different values of $\mu$ and $\epsilon$.
> >
> > ## Our solution
> >
> > We defined the order preserving GDPT. Using the order-preserving property, direct comparisons between $\delta_\mu(\epsilon)$ and $\delta(\epsilon)$ are no longer required; instead, comparing their corresponding GDPTs suffices. Furthermore, by a partial derivative analysis, we picked a special discretization, where the GDPT can be bounded by two stair functions, and solve the comparison problem near the origin. This numerical solution is carefully designed to avoid losing tractability.
> >
> > With Theorem 4.6, we can verify $(\epsilon_h,\mu)$-head GDP conditions for arbitrarily large $\epsilon_h$ and an arbitrarily precise approximation of $\mu$. Then we discussed the final gap of only being able to verify GDPT up to a finite $\epsilon_h$. We conclude that the gap won't make any notable difference in practice with a proper choice of $\mu$ and $\epsilon_h$. In our example, the additional risk is less than accepting a $10^{-202}$ probability of failure that can also be entirely eliminated with a few extra steps (Theorem 4.7).

---

> > ### Comment · Reviewer_H1fA · 2021-12-01
> > **Re Reply to reviewer H1fa**
> >
> > I'd like to thank the authors for the reply. I think I understand better now the purpose and usefulness of GDPT, and hence have raised the score accordingly. However, my other concern remains: more evaluations are needed to demonstrate the usefulness of this method (and compare with other non-GDP methods).

---

> > > ### Author Response · Authors · 2021-12-07
> > > **Reply for H1fa**
> > >
> > > Thank you for your valuable suggestions and consideration.
> > >
> > > As for your concern of evaluations, the goal of our work is to bridge the gap between old algorithms and the framework of GDP. The analysis about the GDP itself is beyond the scope of this paper. For more detailed discussion of GDP itself, we encourage readers to refer to [1].
> > >
> > > [1] - Jinshuo Dong, Aaron Roth, and Weijie J Su. Gaussian differential privacy.To appear in Journal of the Royal Statistical Society: Series B (Statistical Methodology), 2019.

---

### Official Review · Reviewer_RPM2 · 2021-11-02

**Correctness:** 3
**Technical Novelty And Significance:** 2
**Empirical Novelty And Significance:** Not applicable
**Recommendation:** 5
**Confidence:** 4

**Main Review:**

The work presents a comprehensive overview of the GDP in the context of privacy profiling, that is attempting to reduce the detrimental effects of DP on the results of algorithmic training.

There are many interesting insights presented based on the results of a closer analysis of GDP in regards to its behaviour around the head/tail composition of the function through the lens of the novel formulation of GDPT (including experimentally shown tighter privacy analysis, resulting in better utility of the algorithm).

The motivation is very clear to me, as the existing formulations of DP are notoriously difficult to interpret by a non-specialist, require multiple parameters that are not easily defined in practise and provide guarantees that can be difficult to put into perspective. In addition, DP is known to be detrimental to the utility of the trained model, which a more carefully selected formulation (such as GDP) can partially mitigate. Therefore, I believe that this work is a needed step towards democratisation of privacy-preserving ML and a timely investigation into mitigations of DP’s detrimental effects to algorithmic training performance.

Major comments:

While the motivation behind the analysis of GDP is clear, the potential applications of the work (in general) are too, the take-home message of the findings is not. Is the key message of the paper that GDP gives tighter guarantees? Or that existing DP formulations are pessimistic and detrimental to utility? It is very difficult to assess the impact of the work, when there is no clear message on how and why it needs to be integrated into the existing workflows. To alleviate some of these issues I would suggest being much more explicit in the conclusion and contributions: in my eyes GDPT on its own is difficult to put into practical context. I do, however, acknowledge that Section 5 introduces how GDPT can be applied to existing notions, but in my opinion, this comes very late and the conclusions are very ad-hoc. There is no clear direction of how the insights from this work are meant to be interpreted and what the main message of the manuscript is.

In terms of content, my main concern here is the overall novelty of the work and the insights presented. There exists a similar line of work on the interpretation of GDP by Asoodeh et al. [1], that was published in March 2021 and presents an argument about the properties of GDP along with its relation to DP, and the associated effects on the privacy-utility trade-off. This work, however, was not cited or addressed in this manuscript. Similar things can be said about Bu et al.[2], which considers the application of GDP in the context of deep learning. Many of the insights seen in this work (e.g. the interpretation of GDP and its relation to DP in general) have already been presented in these two works and are thus hardly novel. While the goals of this work and [1] are tangential, and GDPT being based on DP analysis through the head/tail behaviour and not on a reinterpretation of other DP formulations and authors here consider Laplace mechanism rather than Gaussian, the conclusions these works present are very similar.
I would like, therefore, to request that reviewers compare their contributions to the work by [1] and specify the novel conclusions that can be drawn from their work in comparison to the prior work. I am open to changing my review should the authors present a sound argument that their work derives additional insights when compared to the literature I linked and particularly, their implications for the wider scientific community.

[1] - Asoodeh, Shahab, et al. "Three variants of differential privacy: Lossless conversion and applications." IEEE Journal on Selected Areas in Information Theory 2.1 (2021): 208-222.
[2] - Bu, Zhiqi, et al. "Deep learning with Gaussian differential privacy." Harvard data science review 2020.23 (2020).

Another concern I have is a lack of any discussion on the Gaussian mechanism. As this paper is positioned to be applicable for ML practitioners, I find the scope of these findings rather limiting (especially if we were to consider deep learning) if only the Laplacian mechanism is considered due to its unfavourable composition properties. Moreover, I am unsure if it is enough to only assess Laplacian in the work that considers Gaussian DP and completely disregard the Gaussian mechanism. In general, I would like to see justification on why Laplace is the suitable choice here, as otherwise I am struggling to see the broader impact of this work and would, therefore, encourage authors to include some concrete applications of their work.

On this note, I would have expected a more detailed discussion on the applications of this work (or of DP in general) in the domain of ML specifically. I would encourage authors to expand their Section 5 and provide explicit applications of their work in this domain.

The experimental results (Figure 1) could be explained better. The results are relatively straightforward, but the subsection comes unannounced and following the method of obtaining these was not straightforward.

Minor comments:
The actual novelty of the paper is not very clearly presented, the abstract contains a lot of information about GDPT’s details, so the actual contributions are diluted (and the same thing can be said about the introduction).  As a result, it is very difficult to determine the actual novel contributions that the work contains. I would suggest trimming the technical details down to make the contributions of the paper more convincing.

Multiple issues with Definition 1.1:
 missing a less than or equals sign.
 does not specify the condition of ‘differ’: is this an add/remove a record or is it a replace a record?


**Summary Of The Paper:**

This work is an investigation into the newly proposed Gaussian Differential privacy (GDP) in the context of existing formulations of epsilon, delta differential privacy (DP). Authors propose a novel framework for GDP transformation (GDPT) that analyses the properties of GDP and contrasts it to DP through the lens of the adversarial (head/tail composition) analysis, showing both theoretically and empirically that it results in better algorithmic utility. Authors provide a list of applications for GDPT and outline the benefits of this approach (primarily through tighter privacy bounds).


**Summary Of The Review:**

Overall, while I do believe that this paper tries to address a very important problem and is formally sound, I am not fully convinced that there is enough novelty in the existing manuscript to justify the acceptance and therefore I believe that this paper is just below the bar of acceptance.

---

> ### Author Response · Authors · 2021-11-22
> **Reply to reviewer RPM2**
>
>
> We sincerely appreciate your time and effort in reviewing our paper. Below, we address all your concerns. Please let us know if there are any further questions.
>
> ## 1. Comparison and novelty
>
> First, we want to compare our work with [1] and [2].
>
> In [1], the authors developed a machinery to symbolically lossless translate between DP and RDP as well as f-DP and RDP.
> Combined with the existing lossless link between DP and f-DP, they completed the framework of lossless conversion between those three notations.
>
> From our perspective, the [2] is a concise version of Dong et  al.(2019) that based on a deep-learning instead of a statistical point of view. Without explicit usage of, they developed novel SGD algorithms with the insights from GDP notations. The lossless conversion also appeared in this paper as (6).
>
> The problem we consider is different. The starting point of our work is the Corollary 2.13 (Theorem 3.2 is our paper) we cite from [3]. This corollary itself is the lossless conversion between DP and f-DP. We consider what to do "after" the lossless conversion is established. On page 4 of our work, after a restatement of the lossless conversion result, we pointed out that this theoretically perfect link does not automatically lead to a meaningful way to identify GDP algorithms because the numerical evaluation is either impossible or at an affordable cost of tractability. Our paper bridged this gap. We extracted Example 2 from our paper with additional comments on how our new tools fix what Theorem 3.2 is lacking and we will show this in a separate comment due to length limitations.
>
> ## 2. No discussion on the Gaussian mechanism
>
> Paradoxically, the Gaussian mechanism is the only mechanism that our methods are definitely not applicable on. The GDP condition is an abstraction of DP guarantee of the Gaussian mechanism, and therefore, the DP guarantee of Gaussian mechanisms is already in the form of GDP (Theorem 2.7 in [3]).
>
> Aside from providing more insights about GDP, the main goal of our paper is to build a tractable bridge between DP and GDP. The motivations behind this goal are the advantages of GDP itself and the rich literature and tools developed using the notation of DP. We treat the mechanisms as black boxes and only work on privacy guarantees that are known to us. One may argue that opening the black boxes and redesigning the mechanisms from scratch will lead to be better outcomes. However, we want to point out that redesigning the mechanisms from scratch is not always possible, especially when the data has already been collected and perturbed.
>
> ## 3. Too much GDPT’s details, so the actual contributions are diluted
>
> In chapter 5, we explored some possibilities for other ways of using GDPT. However, we considered the GDPT's details (Chapter 4) as actual contributions of our work. The GDPT is an easy and tractable way of converting from DP to GDP. With the help of GDPT, we solved the identification and measurement problems. Algorithms that are not developed under the GDP framework can therefore be incorporated into the framework of GDP with minimum effort.
>
> ## 4. The Fig. 1 could be explained better
>
> The left side of Fig. 1 compares the unrefined privacy profile (naively derived from the tradeoff function) and the refined privacy profile. The unrefined curve and refined curve only differ near the origin. When the curve is unrefined, the value converge to $1$ for example 2 (SGD) and example 3 (ICEA).
>
> ## 5. Other Minor comments
>
> •   The missing sign problem is fixed.  Thank you for pointting out.
>
> •   We didn't choose the exact meaning of "differ" because the exact choice won't affect the link between GDP and privacy profiles as long as the same definition is used on both sides. However, the results of sub-sampling are based on the "add/remove one" definition of "differ". We clarified that in 5.3.
>
> ## 6. Reference
>
> [1] - Asoodeh, Shahab, et al. "Three variants of differential privacy: Lossless conversion and applications." IEEE Journal on Selected Areas in Information Theory 2.1 (2021): 208-222.
>
> [2] - Bu, Zhiqi, et al. "Deep learning with Gaussian differential privacy." Harvard data science review 2020.23 (2020).
>
> [3] - Jinshuo Dong, Aaron Roth, and Weijie J Su.  Gaussian differential privacy.To appear in Journal of the Royal Statistical Society: Series B (Statistical Methodology), 2019.

---

> > ### Author Response · Authors · 2021-11-22
> > **Example**
> >
> > # Example
> > In this subsection, we extracted Example 2 from our paper with additional comments on how our new tools fix what previous works are lacking.
> >
> > Theorem 3.2 (Theoretical conversion): A mechanism is $\mu$-GDP if and only if it is $(\epsilon, \delta_\mu(\epsilon))$-DP for all $\epsilon \geq 0$, where
> >
> >
> > $$\delta_\mu(\epsilon)=\Phi\left(-\frac{\epsilon}{\mu}+\frac{\mu}{2}\right)-\mathrm{e}^{\epsilon} \Phi\left(-\frac{\epsilon}{\mu}-\frac{\mu}{2}\right) .$$
> >
> >
> > In the example 2, suppose we are given an algorithm $X$ where the noise parameter has the form $\sigma=A\epsilon^{-1}\sqrt{\log (B / \delta)}$. For simplicity, we let $\sigma=A=B=1$.
> >
> > We start with the identification problem: is $X$ GDP?
> >
> >
> >
> >
> > ##Issues prior to our work
> > According to Theorem 3.2, $X$ is GDP if and only if the implicit $\delta(\epsilon)$ satisfies $\delta(\epsilon)\leq \delta_\mu(\epsilon)$ for some finite $\mu$. A naive application of Theorem 3.2 will lead to a failure: It is easy to see that $\delta_\mu(\epsilon)<1$ for any $\mu>0$ but $\delta(\epsilon)\geq 1$ for sufficiently small $\epsilon$. This naive comparison will disqualify $X$ as GDP by mistake.
> >
> > ## Our solution:
> >
> > To solve this problem, we give Theorem 3.1, Theorem 3.2 and Corollary 3.2 to show that $\delta(\epsilon)< 1$ as long as $X$ provides a single arbitrarily weak non-trival $(\epsilon,\delta)$-DP guarantee.
> >
> > ## Issues prior to our work:
> >
> >  Even with $\delta(\epsilon)$ correctly specified, the existence of $\mu$ such that $\delta(\epsilon)<\delta_\mu(\epsilon)$ for all $\epsilon\geq 0$ is not easy to decide.
> >
> > ## Our solution
> >
> > As we demonstrated in Theorem 4.3, the existence of $mu$ is solely determined by an asymptotic comparison of $\delta(\epsilon) $and $\delta_\mu(\epsilon) $at $+\infty$.The asymptotic characterization is given as Lemma A.1. Therefore, the solution to the identification problem is wrapped up in a simple theorem (Theorem 4.4).
> >
> > Then we move forward to the measurement problem. As a GDP algorithm, what is the $\mu$ of $X$?
> >
> > ## Issues prior to our work
> >  The Theorem 3.2 gives that $\mu$ is the smallest $\mu_0$ such that $\delta(\epsilon)\leq\delta_{\mu_0}(\epsilon)$ for all $\epsilon\geq 0$. We would be forced to search through an uncountably-large family of functions for a single $\delta_\mu(\epsilon)$ that never crosses $\delta(\epsilon)$ anywhere on $[0,\infty)$ and has $\mu$ as small as possible. Those conditions cannot be check numerically, even for a single $\mu$. Turning to closed forms for privacy profiles and $\delta_\mu$ is also difficult: even if a given privacy profile is easy to handle, $\delta_\mu$ presents some technical hurdles. The profile $\delta_\mu$ and $\Phi$ are transcendental with different asymptotic behaviors for different values of $\mu$ and $\epsilon$.
> >
> > ## Our solution
> >
> > We defined the order preserving GDPT. Using the order-preserving property, direct comparisons between $\delta_\mu(\epsilon)$ and $\delta(\epsilon)$ are no longer required; instead, comparing their corresponding GDPTs suffices. Furthermore, by a partial derivative analysis, we picked a special discretization, where the GDPT can be bounded by two stair functions, and solve the comparison problem near the origin. This numerical solution is carefully designed to avoid losing tractability.
> >
> > With Theorem 4.6, we can verify $(\epsilon_h,\mu)$-head GDP conditions for arbitrarily large $\epsilon_h$ and an arbitrarily precise approximation of $\mu$. Then we discussed the final gap of only being able to verify GDPT up to a finite $\epsilon_h$. We conclude that the gap won't make any notable difference in practice with a proper choice of $\mu$ and $\epsilon_h$. In our example, the additional risk is less than accepting a $10^{-202}$ probability of failure that can also be entirely eliminated with a few extra steps (Theorem 4.7).

---

> > ### Comment · Reviewer_RPM2 · 2021-11-29
> > **Response to a rebuttal**
> >
> > I thank the authors for these clarifications. It is now more clear to me how this work would fit in the context of DP, especially when compared to the works I linked above. While the response clarifies a number of things, and does take into account my issues with how similar some of these insights are to the ones derived in prior works, I am still uncertain if I see a significant enough scientific contribution to recommend acceptance.

---

### Official Review · Reviewer_ZW49 · 2021-11-03

**Correctness:** 4
**Technical Novelty And Significance:** 1
**Empirical Novelty And Significance:** Not applicable
**Recommendation:** 3
**Confidence:** 5

**Main Review:**

This paper attempts to identify a class of DP algorithms they call GDP algorithms. It is largely unclear what is the motive of the paper and what is the application. I will list down some of the issues below:

1. The theorems in section 3 are either results of previous works or are at best trivial facts.

2. The theorems in section 4 are just basic algebraic manipulations. This paper thus lacks technical novelty.

3. The purpose of the introduction of the transform is unclear to me. There is no formal result showing the improvement in utility for a class of algorithms. The abstract claims their technique refines the utility, but this is illustrated using only 1 example and some basic calculations.

4. The abstract claims they study the effect of subsampling on the GDPT, but the only thing mentioned about subsampling is that it doesn't give any amplification.

Largely, I don't see any applicability of the transformation introduced in this paper.

**Summary Of The Paper:**

In this paper, the authors propose the Gaussian Differential Privacy Transformation to identify which algorithms fall under the framework of of GDP.

**Summary Of The Review:**

The paper lacks technical novelty and makes no real contribution/improvement to support the new transformation for characterizing introduced. Thus, I think it is not good enough for publication.

---

> ### Author Response · Authors · 2021-11-22
> **Reply to reviewer ZW49**
>
>
> We thank the reviewer for the feedback. However, we are afraid that there might be some fundamental misunderstanding of our work. Below, we address all your concerns. Please let us know if there are any further questions.
>
> ## This paper attempts to identify a class of DP algorithms they call GDP algorithms.
>
> There might be a serious misunderstanding. The class of GDP is not defined by us, GDP is an established framework of DP purposed in [1]. The advantages of GDP include but are not limited to:
>
>
> + parameterized only by a single value $\mu$,
>
> + easy to interpret and,
>
> + closed under a tight theorem of composition.
>
>
> In addition, as is pointed out in [1], any privacy definition that retains a hypothesis testing interpretation, the privacy guarantee of composition with an appropriate scaling
> converges to GDP in the limit.
>
> [1] - Jinshuo Dong, Aaron Roth, and Weijie J Su.  Gaussian differential privacy.To appear in Journal of the Royal Statistical Society: Series B (Statistical Methodology), 2019.
>
>
>
> ## It is largely unclear what is the motive of the paper and what is the application
>
> The starting point of our work is the Corollary 2.13 (Theorem 3.2 is our paper) we cite from [1]. This corollary itself is the lossless conversion between DP and f-DP. We consider what to do "after" the lossless conversion is established. On page 4 of our work, after a restatement of the lossless conversion result, we pointed out that this theoretically perfect link does not automatically lead to a meaningful way to identify GDP algorithms because the numerical evaluation is either impossible or at an affordable cost of tractability. Our paper bridged this gap. In the following comment, we extracted Example 2 from our paper with additional comments on how our new tools fix what Theorem 3.2 is lacking.
>
> ## The theorems in section 3 are either results of previous works or are at best trivial facts
>
> We respectfully disagree. The partial order is theoretically critical and is commonly overlooked. As is shown in Figure 1 (left), the corresponding refinement is significant for the privacy profile near the origin. Without the refinement, the trade-off function itself cannot correctly specify some algorithms (e.g., example 2 and 3) offers non-trivial $(0,\delta)$-DP guarantees and therefore cannot be correctly classified as GDP.
>
> ## The theorems in section 4 are just basic algebraic manipulations. This paper thus lacks technical novelty
>
> We respectfully disagree. Here are some examples of technological innovation:
>
>
>
> + In Theorem 4.5, we have shown the implicit function linking DP and GDP and Lipschitz (bounded derivative) on $\epsilon$;
>
> + In Theorem 4.6, the forms of staircase functions are carefully picked to make sure the gap between $G^+$ and $G^-$ can be controlled by the constant $D$ in a tractable way;
>
> + In Section 4.2, we give an explicit characterization of the possible privacy loss from only checking a finite interval on the privacy profile;
>
> + In Theorem 4.7, we show that the privacy loss from only checking a finite interval can be entirely eliminated with a few extra steps.

---

> > ### Author Response · Authors · 2021-11-22
> > **Example**
> >
> > # Example
> > In this section, we extracted Example 2 from our paper with additional comments on how our new tools fix what previous works are lacking.
> >
> > Theorem 3.2 (Theoretical conversion): A mechanism is $\mu$-GDP if and only if it is $(\epsilon, \delta_\mu(\epsilon))$-DP for all $\epsilon \geq 0$, where
> >
> >
> > $$\delta_\mu(\epsilon)=\Phi\left(-\frac{\epsilon}{\mu}+\frac{\mu}{2}\right)-\mathrm{e}^{\epsilon} \Phi\left(-\frac{\epsilon}{\mu}-\frac{\mu}{2}\right) .$$
> >
> >
> > In the example 2, suppose we are given an algorithm $X$ where the noise parameter has the form $\sigma=A\epsilon^{-1}\sqrt{\log (B / \delta)}$. For simplicity, we let $\sigma=A=B=1$.
> >
> > We start with the identification problem: is $X$ GDP?
> >
> >
> >
> >
> > ##Issues prior to our work
> > According to Theorem 3.2, $X$ is GDP if and only if the implicit $\delta(\epsilon)$ satisfies $\delta(\epsilon)\leq \delta_\mu(\epsilon)$ for some finite $\mu$. A naive application of Theorem 3.2 will lead to a failure: It is easy to see that $\delta_\mu(\epsilon)<1$ for any $\mu>0$ but $\delta(\epsilon)\geq 1$ for sufficiently small $\epsilon$. This naive comparison will disqualify $X$ as GDP by mistake.
> >
> > ## Our solution:
> >
> > To solve this problem, we give Theorem 3.1, Theorem 3.2 and Corollary 3.2 to show that $\delta(\epsilon)< 1$ as long as $X$ provides a single arbitrarily weak non-trival $(\epsilon,\delta)$-DP guarantee.
> >
> > ## Issues prior to our work:
> >
> >  Even with $\delta(\epsilon)$ correctly specified, the existence of $\mu$ such that $\delta(\epsilon)<\delta_\mu(\epsilon)$ for all $\epsilon\geq 0$ is not easy to decide.
> >
> > ## Our solution
> >
> > As we demonstrated in Theorem 4.3, the existence of $mu$ is solely determined by an asymptotic comparison of $\delta(\epsilon) $and $\delta_\mu(\epsilon) $at $+\infty$.The asymptotic characterization is given as Lemma A.1. Therefore, the solution to the identification problem is wrapped up in a simple theorem (Theorem 4.4).
> >
> > Then we move forward to the measurement problem. As a GDP algorithm, what is the $\mu$ of $X$?
> >
> > ## Issues prior to our work
> >  The Theorem 3.2 gives that $\mu$ is the smallest $\mu_0$ such that $\delta(\epsilon)\leq\delta_{\mu_0}(\epsilon)$ for all $\epsilon\geq 0$. We would be forced to search through an uncountably-large family of functions for a single $\delta_\mu(\epsilon)$ that never crosses $\delta(\epsilon)$ anywhere on $[0,\infty)$ and has $\mu$ as small as possible. Those conditions cannot be check numerically, even for a single $\mu$. Turning to closed forms for privacy profiles and $\delta_\mu$ is also difficult: even if a given privacy profile is easy to handle, $\delta_\mu$ presents some technical hurdles. The profile $\delta_\mu$ and $\Phi$ are transcendental with different asymptotic behaviors for different values of $\mu$ and $\epsilon$.
> >
> > ## Our solution
> >
> > We defined the order preserving GDPT. Using the order-preserving property, direct comparisons between $\delta_\mu(\epsilon)$ and $\delta(\epsilon)$ are no longer required; instead, comparing their corresponding GDPTs suffices. Furthermore, by a partial derivative analysis, we picked a special discretization, where the GDPT can be bounded by two stair functions, and solve the comparison problem near the origin. This numerical solution is carefully designed to avoid losing tractability.
> >
> > With Theorem 4.6, we can verify $(\epsilon_h,\mu)$-head GDP conditions for arbitrarily large $\epsilon_h$ and an arbitrarily precise approximation of $\mu$. Then we discussed the final gap of only being able to verify GDPT up to a finite $\epsilon_h$. We conclude that the gap won't make any notable difference in practice with a proper choice of $\mu$ and $\epsilon_h$. In our example, the additional risk is less than accepting a $10^{-202}$ probability of failure that can also be entirely eliminated with a few extra steps (Theorem 4.7).

---

### Official Review · Reviewer_qUya · 2021-11-03

**Correctness:** 4
**Technical Novelty And Significance:** 4
**Empirical Novelty And Significance:** Not applicable
**Recommendation:** 6
**Confidence:** 3

**Main Review:**

GDP is an elegant framework for analyzing private algorithms, due to its nice composition properties. A wider application of the framework, however, would benefit from automated tools for deriving tight GDP guarantees for a given algorithm. The main strength of the paper is in giving one such tool, and showing how it can ease the privacy analysis of several algorithms. The visualizations of the privacy profiles transformed by GDPT in the paper provide some insight on the privacy guaranteed by several basic algorithm. The main weakness of the paper for me is in the strength of the applications mentioned at the end:
* the analysis of the Laplace noise mechanism is simple, and probably not the best illustration of the power of the methods here;
* the analysis of SGD is limited to giving guarantees for some constant $\varepsilon$ and $\delta$, in particular, a value of $\delta$ that is too high for deployment; the illustration that the partial order on $(\varepsilon, \delta)$ can help improve an algorithm's error analysis is interesting, however;
* I am not sure what the takeaway is from the analysis of subsampling?

So GDPT may end being a useful tool in a privacy toolbox built around GDP, but the paper does not quite manage to make a strong case for it. Because of this, I will be ok if the paper were accepted to ICLR, but I will not push for it.

Some further minor comments:
* DP definition has a typo - missing less than or equal to sign.
* Concentrated DP was first proposed by Dwork and Rothblum (https://arxiv.org/abs/1603.01887).
* The references for DP SGD should contain Bassily, Smith, Thakurta, FOCS 2014
* Is $\mu_{\text{GDP}}(x,y)$ well-defined: what about existence and uniqueness of $\mu$?

**Summary Of The Paper:**

The paper develops tools for analyzing private algorithms in the framework of Gaussian Differential Privacy (GDP). GDP is a strengthening of approximate differential privacy, similar in spirit to concentrated differential privacy (CDP): both GDP and CDP capture the privacy guarantees of the Gaussian noise mechanism, but generalize them in different directions. GDP has a number of nice analytical properties but analyzing the GDP guarantees of an algorithm can be difficult, as it involves comparing two different functions on $[0,\infty)$. The paper proposes a method to reduce this problem to comparing a function against a fixed constant function, making numerical or analytical GDP analysis of a given algorithm easier.

**Summary Of The Review:**

The paper makes a contribution to automating the privacy analysis of algorithms within the elegant GDP framework, but doesn’t manage to make a strong case for the usefulness and applicability of the new tool it develops.

---

> ### Author Response · Authors · 2021-11-22
> **Reply to reviewer qUya**
>
> We sincerely appreciate your time and effort in reviewing our paper. Below, we address all your concerns. Please let us know if there are any further questions.
>
> ## The analysis of the Laplace noise mechanism is simple, and probably not the best illustration of the power of the methods here
>
> The analysis of the Laplace noise mechanism is a simple manifestation of one of the applicability of our methods. The main application are the identifications and measurements throughout chapter 2 to 5. The starting point of our work is the Corollary 2.13 (Theorem 3.2 is our paper) we cite from [1]. We consider what to do "after" the Corollary 2.13 is established. In page 4 of our work, we pointed out that this theoretically perfect link does not automatically lead to a meaningful way to identify GDP algorithms because the numerical evaluation is either impossible or at an unaffordable cost of tractability. Our paper mended this gap. In a separate reply, we extracted the Example 2 from our paper with additional comment on how our new tools fixed what Theorem 3.2 is lacking.
>
> [1] - Jinshuo Dong, Aaron Roth, and Weijie J Su. Gaussian differential privacy.To appear in Journal of the Royal Statistical Society: Series B (Statistical Methodology), 2019.
>
> ## The value of $\delta$ that is too high for deployment
>
> The overlooked partial order is theoretically critical. As is shown on Figure 1 (left), the corresponding refinement is significant for the privacy profile near the origin. Without the refinement, the trade-off function itself cannot correctly specify some algorithms (e.g., example 2 and 3) offers non-trivial $(0,\delta)$-DP guarantee and therefore cannot be correctly classified as GDP. We agree that the improvement shown by the illustration is only an interesting side-effect of the theoretical development. We will clarify this in 5.2.
>
> ## What the takeaway is from the analysis of subsampling?
>
> One insight we found by subsampling is we cannot expect adding a subsampling procedure prior to a $\mu$-GDP algorithm to lead to another GDP algorithm with an improved $\mu$. This is surprising since, under the original DP notation, such improvement was guaranteed (e.g., a 50\% subsampled 1-DP algorithm would be approximately 0.27-DP). One possible interpretation of this result is that subsampling can be used as a building block of GDP algorithms but shouldn't be used as a preprocessing procedure to enhance a GDP algorithm.
>
> ## Minor comments:
>
>
> •   DP definition has a typo - missing less than or equal to sign.
>
> This typo is now fixed.
>
> •   Concentrated DP was first proposed by Dwork and Rothblum.
>
> This reference is now added.
>
> •   The references for DP SGD should contain Bassily, Smith, Thakurta, FOCS 2014
>
> This reference is now added.
>
> •   Is $\mu_{GDP}(x,y)$ well-defined: what about existence and uniqueness of $\mu$?
>
> Yes. Given a fixed $\mu$, $\delta_\mu(\epsilon)$ is a strictly decreasing continuous function of $\epsilon$. Given a fixed $\epsilon$, $\delta_\mu(\epsilon)$ is a strictly increasing continuous function of $\mu$. Therefore, this implicit function is well defined.

---

> > ### Author Response · Authors · 2021-11-22
> > **Example**
> >
> > # Example
> > In this section, we extracted Example 2 from our paper with additional comments on how our new tools fix what previous works are lacking.
> >
> > Theorem 3.2 (Theoretical conversion): A mechanism is $\mu$-GDP if and only if it is $(\epsilon, \delta_\mu(\epsilon))$-DP for all $\epsilon \geq 0$, where
> >
> >
> > $$\delta_\mu(\epsilon)=\Phi\left(-\frac{\epsilon}{\mu}+\frac{\mu}{2}\right)-\mathrm{e}^{\epsilon} \Phi\left(-\frac{\epsilon}{\mu}-\frac{\mu}{2}\right) .$$
> >
> >
> > In the example 2, suppose we are given an algorithm $X$ where the noise parameter has the form $\sigma=A\epsilon^{-1}\sqrt{\log (B / \delta)}$. For simplicity, we let $\sigma=A=B=1$.
> >
> > We start with the identification problem: is $X$ GDP?
> >
> >
> >
> >
> > ##Issues prior to our work
> > According to Theorem 3.2, $X$ is GDP if and only if the implicit $\delta(\epsilon)$ satisfies $\delta(\epsilon)\leq \delta_\mu(\epsilon)$ for some finite $\mu$. A naive application of Theorem 3.2 will lead to a failure: It is easy to see that $\delta_\mu(\epsilon)<1$ for any $\mu>0$ but $\delta(\epsilon)\geq 1$ for sufficiently small $\epsilon$. This naive comparison will disqualify $X$ as GDP by mistake.
> >
> > ## Our solution:
> >
> > To solve this problem, we give Theorem 3.1, Theorem 3.2 and Corollary 3.2 to show that $\delta(\epsilon)< 1$ as long as $X$ provides a single arbitrarily weak non-trival $(\epsilon,\delta)$-DP guarantee.
> >
> > ## Issues prior to our work:
> >
> >  Even with $\delta(\epsilon)$ correctly specified, the existence of $\mu$ such that $\delta(\epsilon)<\delta_\mu(\epsilon)$ for all $\epsilon\geq 0$ is not easy to decide.
> >
> > ## Our solution
> >
> > As we demonstrated in Theorem 4.3, the existence of $mu$ is solely determined by an asymptotic comparison of $\delta(\epsilon) $and $\delta_\mu(\epsilon) $at $+\infty$.The asymptotic characterization is given as Lemma A.1. Therefore, the solution to the identification problem is wrapped up in a simple theorem (Theorem 4.4).
> >
> > Then we move forward to the measurement problem. As a GDP algorithm, what is the $\mu$ of $X$?
> >
> > ## Issues prior to our work
> >  The Theorem 3.2 gives that $\mu$ is the smallest $\mu_0$ such that $\delta(\epsilon)\leq\delta_{\mu_0}(\epsilon)$ for all $\epsilon\geq 0$. We would be forced to search through an uncountably-large family of functions for a single $\delta_\mu(\epsilon)$ that never crosses $\delta(\epsilon)$ anywhere on $[0,\infty)$ and has $\mu$ as small as possible. Those conditions cannot be check numerically, even for a single $\mu$. Turning to closed forms for privacy profiles and $\delta_\mu$ is also difficult: even if a given privacy profile is easy to handle, $\delta_\mu$ presents some technical hurdles. The profile $\delta_\mu$ and $\Phi$ are transcendental with different asymptotic behaviors for different values of $\mu$ and $\epsilon$.
> >
> > ## Our solution
> >
> > We defined the order preserving GDPT. Using the order-preserving property, direct comparisons between $\delta_\mu(\epsilon)$ and $\delta(\epsilon)$ are no longer required; instead, comparing their corresponding GDPTs suffices. Furthermore, by a partial derivative analysis, we picked a special discretization, where the GDPT can be bounded by two stair functions, and solve the comparison problem near the origin. This numerical solution is carefully designed to avoid losing tractability.
> >
> > With Theorem 4.6, we can verify $(\epsilon_h,\mu)$-head GDP conditions for arbitrarily large $\epsilon_h$ and an arbitrarily precise approximation of $\mu$. Then we discussed the final gap of only being able to verify GDPT up to a finite $\epsilon_h$. We conclude that the gap won't make any notable difference in practice with a proper choice of $\mu$ and $\epsilon_h$. In our example, the additional risk is less than accepting a $10^{-202}$ probability of failure that can also be entirely eliminated with a few extra steps (Theorem 4.7).

---

> > ### Comment · Reviewer_qUya · 2021-11-30
> > **Ack**
> >
> > Thank you to the authors for this thorough and detailed response. I am afraid I still don't see a strong enough application of these new tools that would illustrate their value in analyzing algorithms in the GDP framework. The three applications listed in Section 5 are:
> > * the analysis of the Laplace mechanism, which we agree does not show the power of the methods;
> > * the analysis of an $(\varepsilon, \delta)$-DP algorithm for $\delta > 0.067$, a rather large value;
> > * the analysis of subsampling, showing that subsampling does not really GDP; I disagree that this is surprising, since it's well-known for the closely related notion of concentrated differential privacy.
> >
> > I don't know what else the authors mean by "the identifications and measurements throughout chapter 2 to 5". The plots in those sections? These visualisations are nice visual aids to the discussion in the paper, but they need to be developed further to constitute strong enough applications of the new methods. For example, can the authors show a much tighter GDP analysis of something like SGD, leading to an improved utility-privacy tradeoff compared to existing analyses?

---

> > > ### Author Response · Authors · 2021-12-07
> > > **Reply To qUya**
> > >
> > > Thank you for your reply.
> > >
> > > Please refer to the example for "the identifications and measurements throughout chapter 2 to 5".
> > >
> > > For an algorithm M, it is natural to ask, "is M GDP?". Our identification procedure answers this question by taking the limit of the privacy profile.
> > >
> > > For an algorithm that is known to be GDP, the logical next step is find the $\mu$. Our measurements procedure answers this question by computing two stair functions.
> > >
> > > The tightness of GDP is beyond the scope of our paper and is already discussed in [1].
> > >
> > > [1] - Jinshuo Dong, Aaron Roth, and Weijie J Su.  Gaussian differential privacy.To appear in Journal of the Royal Statistical Society: Series B (Statistical Methodology), 2019.

---

### Decision · Program_Chairs · 2022-01-20

**Decision:**

Reject

**Comment:**

We thank the authors for their response. The reviewers agree that this paper provides contributions in automating privacy analyses under the Gaussian differential privacy (GDP) framework. The reviewers also pointed out several drawbacks of the paper. Most importantly, the reviewers do not find the presented applications to be convincing. In particular, the presented result can be much strengthened if the proposed method can lead to improved privacy analysis for more sophisticated algorithms such as DP-SGD across a wide regime of epsilon and delta. (In general, the privacy guarantee is very weak with delta bigger than 1/n.) Overall, the paper does not seem to provide enough evidence to showcase the usefulness of their proposed method.